# Geometry and topology tango in ordered and amorphous chiral matter

**M. Guzmán⋆, D. Bartolo and D. Carpentier**

ENS de Lyon, CNRS, Laboratoire de Physique, Lyon, France

⋆ marcelo.guzman-jara@ens-lyon.fr

## Abstract

Systems as diverse as mechanical structures and photonic metamaterials enjoy a common geometrical feature: a sublattice or chiral symmetry first introduced to characterize electronic insulators. We show how a real-space observable, the chiral polarization, distinguishes chiral insulators from one another and resolve long-standing ambiguities in the very concept of their bulk-boundary correspondence. We use it to lay out generic geometrical rules to engineer topologically distinct phases, and design zero-energy topological boundary modes in both crystalline and amorphous metamaterials.

Copyright M. Guzmán *et al.*.
This work is licensed under the Creative Commons
Attribution 4.0 International License.
Published by the SciPost Foundation.

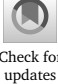

# 1 Introduction

A century after the foundations of band theory in solids by Félix Bloch [1], physicists have discovered new states of electronic matter ranging from insulators to superconductors by exploiting the topological structure of Bloch theory [2–7]. This topological revolution has built on two cornerstones: an abstract classification based on symmetries [8–15], and the practical correspondence between bulk topology and the boundary states measurable in experiments [2–6, 16–19]. During the past decade, these two generic principles spanned fields as diverse as photonics, acoustics, or mechanics, leading to design principles and practical realizations of maximally robust waveguides [20, 21].

Among the number of symmetries constraining wave topology, chiral symmetry has a special status. Out of the three fundamental symmetries of the overarching ten-fold classification [8–10], it is the only one naturally realized with both quantum and classical waves. It generically takes the form of a sub-lattice symmetry when waves propagate in frames composed of two connected lattices $A$ and $B$, with couplings only between, $A$ and $B$ sites, see e.g. Fig. 1a. In electronic systems, the archetypal example of a chiral insulator is provided by the polyacetylene molecule described by the Su-Schrieffer-Hegger (SSH) model [22]. In mechanics, the Hamiltonian description of bead-and-spring networks is intrinsically chiral [23–26]: the $A$ sites correspond to the beads, and the $B$ sites to the springs. In topological photonics and cold atoms chiral wave guides are among the simplest realizations of topological phases. Over the past decade, the modern theory of electronic polarization based on Zak phases and non-Abelian Wilson loops [27–30] has illuminated the intimate relation between crystalline symmetries and the topology of band structures [11–15]. By contrast, the role of chiral symmetry has been overlooked.

In this article, by introducing the concept of chiral polarization we determine the zero-mode content of interfaces between topologically incompatible crystalline and amorphous chiral (meta)materials.

In the bulk, the chiral charge, which measures the imbalance between the number of sites on the sub-frames $A$ and $B$, predicts the number of zero-energy modes of all Hamiltonians defined on a given chiral frame. To characterize chiral insulators we define their chiral polarization $\mathbf{\Pi}$ as the spatial imbalance of the bulk waves on the two sub-frames. This material property does not rely on any crystalline symmetry and can therefore be defined locally on disordered and amorphous frames. In crystals, although akin to the time-reversal polarization of $\mathbb{Z}_2$ insulators [31], we show that $\mathbf{\Pi}$ is not merely set by the Bloch-Hamiltonian topology but also by the underlying frame geometry. At boundaries, we show how $\mathbf{\Pi}$ prescribes the surface chiral charge, and therefore the full zero-energy edge content of crystalline and amorphous chiral matter. Finally, we propose a series of practical protocols to experimentally measure the chiral polarization of mechanical, and photonic chiral metamaterials.

## 2 From chiral charge to chiral polarization and Zak phases

Introducing the concepts of chiral charge and polarization, we demonstrate that bulk properties of chiral matter are determined by an interplay between the frame topology, the frame geometry and the chiral Zak phases of Bloch Hamiltonians.

### 2.1 Chiral charge and chiral polarization.

We consider the propagation of waves in chiral material associated to $d$-dimensional frames including two sub-frames $A$ and $B$. The wave dynamics is defined by a Hamiltonian $\mathcal{H}$. By definition, the chiral symmetry translates in the anti-commutation of $\mathcal{H}$ with the chiral unitary operator $\mathbb{C} = \mathbb{P}^A - \mathbb{P}^B$, where $\mathbb{P}^A$ and $\mathbb{P}^B$ are the two orthogonal projectors on the sub-frames $A$ and $B$. Simply put, in the chiral basis where $\mathbb{C}$ is diagonal, $\mathcal{H}$ is block off-diagonal.

In order to determine the relative weight of the wave functions of $\mathcal{H}$ on the two sub-frames, we introduce the chiral charge

$$\mathcal{M} = \langle \mathbb{C} \rangle, \tag{1}$$

where the average is taken over the complete Hilbert space. Using the basis of fully localized states, we readily find that $\mathcal{M}$ is fully prescribed by the frame topology: the chiral charge counts the imbalance between the number of $A$ and $B$ sites: $\mathcal{M} = N^A - N^B$ . We can however also evaluate Eq. (1) in the eigenbasis of $\mathcal{H}$. Indexing by $n$ the eigenenergies of $\mathcal{H}$, the eigenstates of the chiral Hamiltonian come by pairs of opposite energies related by $|-n\rangle = \mathbb{C} |n\rangle$. Chirality therefore implies that the chiral charge is solely determined by the zero modes of $\mathcal{H}$ as $\mathcal{M} = \sum_n \langle n| \mathbb{C} |n\rangle = \langle 0| \mathbb{C} |0\rangle$. Noting that the $|0\rangle$ states are eigenstates of the chiral operator with eigenvalue $+1$ when localized on the $A$ sites and $-1$ when localized on the B sites, it follows that $\mathcal{M}$ also is an algebraic count the zero modes of $\mathcal{H}$:

$$\mathcal{M} = N^A - N^B = \nu^A - \nu^B. \tag{2}$$

This equality is the classical result established by Maxwell and Calladine in the context of structural mechanics [33, 34] and independently discussed by Sutherland in the context of electron localization [35]. Eq. (2) implies that the spectral properties of $\mathcal{H}$ are constrained by the frame topology. In particular, frames with a non-vanishing chiral charge impose *all* chiral Hamiltonian to possess flat bands. This simple prediction is illustrated in Fig. 1 where we show the Lieb and the dice lattices, which are both characterized by a unit chiral charge per unit

**a.**

**b.**

Figure 1: **Lattices with a finite chiral charge**. **a.** The Lieb (left) and dice (right) frames are both characterized by an imbalance between the number $N^A$ and $N^B$ of sites. In both cases the chiral charge per unit cell equals 1. Any Hamiltonian defined on these frames possesses a flat energy band. **b.** Illustration of two band spectra associated to chiral Hamiltonians defined on the Lieb (left) and dice (right) frames. The two band spectra are computed for tight-binding Hamiltonians with nearest neighbour coupling and a hopping parameter set to 1, see e.g. [32].

cell. All Hamiltonians defined on these lattices are therefore bound to support at least one flat band, Fig. 1b. No chiral insulators exist on the Lieb and dice lattices.

By contrast, in chiral insulators, no zero-energy bulk modes exist and $\mathcal{M}$ must vanish. To probe the relative weight of the wave functions on the two sub-frames, we therefore introduce the chiral polarization vector $\Pi_j = \langle \mathbb{C} x_j \rangle_{E \neq 0}$. As the $|\pm n\rangle$ states contribute equally to $\Pi$ in chiral systems, we henceforth use the definition

$$\Pi_j = 2\langle \mathbb{C} x_j \rangle_{E<0}, \tag{3}$$

with $j = 1, \ldots, d$ are the indices of the $d$ crystallographic directions and where $E < 0$ indicates that the average is taken over the occupied states. This definition differs from the skew polarization introduced in [36,37] for topological insulators, and the mean chiral displacement of

quantum walks [38]. $\Pi_j$ does not rely on any Bloch representation and is therefore defined also in amorphous phases. We stress that, even in the crystalline case, $\Pi_j$ includes geometrical content absent from the skew polarization, as it resolves the weighted positions with a sub-unit-cell resolution. This difference is simply explained by considering the mean chiral displacement (MCD) defined on a 1D lattice given a definition of a unit cell. It is defined per wavepacket $\psi$ as MCD $= \langle\psi|\mathbb{C}x_{\text{UC}}|\psi\rangle$, with $x_{\text{UC}}$ being the position operator at the scale of the unit cell. $x_{\text{UC}}$ defines the positions as integer multiples of the length $a$ of the unit cell. In contrast, equation (3) depends on the actual position of the sites: $x = x_{\text{UC}} + \delta x$, where $\delta x$ is a sublattice correction to the unit-cell position . These differences are not mere technicalities, and will prove crucial in the next sections.

To gain more physical insight, it may be worth noting that in electronic systems, $\Pi_j$ corresponds to the algebraic distance between the charge centers associated to the $A$ and $B$ atoms. While in mechanical networks, $\Pi_j$ is the vector connecting the stress-weighted and displacement-weighted positions. A vanishing polarization indicates that the average locations of the stress and displacement coincide. Conversely, a finite chiral polarization reveals an asymmetric mechanical response discussed in [39,40]. For the sake of clarity, before revealing topologically protected zero modes in amorphous phases, we first explore the consequences of a finite chiral polarization in periodic systems such as in the paradigmatic example of the SSH model illustrated Fig. 2.

## 2.2 Chiral polarization: an interplay between Zak phases and frame geometry.

We begin with a thorough discussion of crystalline materials, defined by periodic frames and Bloch Hamiltonians. Building on previous works on the electronic polarization [27–30],

we relate the chiral polarization of a crystalline material to the two Zak phases of waves projected on sub-lattices $A$ and $B$ when transported across the Brillouin zone. To do so, we first choose a unit cell and consider the basis of Bloch states $|\boldsymbol{k},\alpha\rangle = \sum_{\boldsymbol{R}} e^{i\boldsymbol{k}\cdot\boldsymbol{R}}|\boldsymbol{R}+\boldsymbol{r}_\alpha\rangle$, where $\boldsymbol{R}$ is a Bravais lattice vector, $\alpha$ labels the atoms in the unit cell and $\boldsymbol{k}$ is the momentum in the Brillouin Zone (BZ). We henceforth use a convention where the Bloch Hamiltonian $H(\boldsymbol{k})$ is periodic in the BZ, see [28,41] and Appendix A. More quantitatively, considering first Hamiltonians with no band crossing [1], we define the $A$ sub-lattice Zak phase of the $n^{\text{th}}$ energy band along the crystallographic direction $j$ as

$$\gamma_j^A(n) = i \int_{\mathcal{C}_j} d\boldsymbol{k} \, \langle u_n|\mathbb{P}^A \partial_{\boldsymbol{k}} \mathbb{P}^A|u_n\rangle \,, \tag{4}$$

where the $|u_n(\mathbf{k})\rangle$ are the eigenstates of $H(\boldsymbol{k})$, and $\mathcal{C}_j$ the non-contractible loops over the Brillouin zone defined along the $d$ crystallographic axes. $\gamma_j^B(n)$ is defined analogously on the $B$ sublattice. The (intercellular) Zak phase is given by the sum of $\gamma_j^A(n)$ and $\gamma_j^B(n)$ [43]. In Appendix B, we show how to decompose the chiral polarization into a spectral and a frame contribution:

$$\Pi_j = \frac{a}{\pi}(\gamma_j^A - \gamma_j^B) + p_j \,, \tag{5}$$

where $a$ is the lattice spacing (assumed identical in all directions), $\gamma_j^A$ and $\gamma_j^B$ are the sublattice Zak phases defined by

$$\gamma_j^A = \sum_{n<0} \gamma_j^A(n). \tag{6}$$

---

[1]In the situation where bands cross, our results should be generalized resorting to the Wilson loops of the non-commutative Berry connexion instead of the abelian Zak phase connection [42].

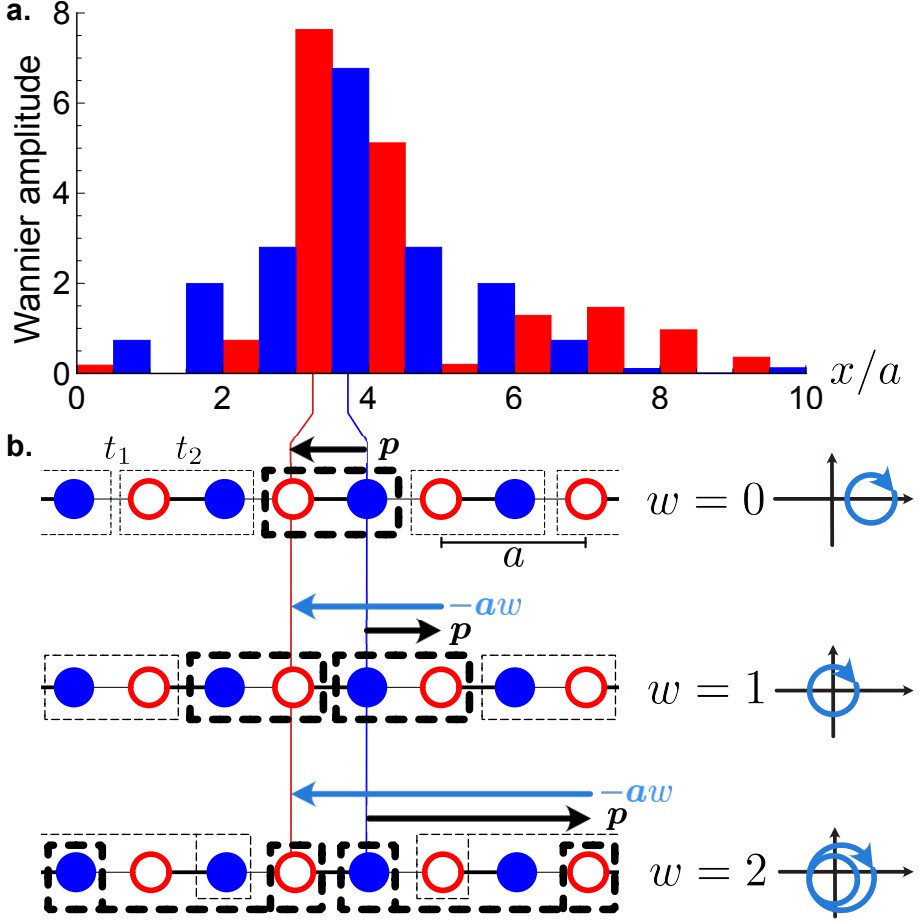

Figure 2: **Chiral polarization and Wannier functions. a.** Square of the Wannier amplitude projected into the $A$ (red) and $B$ (blue) sublattices for the ground state configuration of the two-band SSH model with hopping ratio $t_1/t_2 = 0.79$. $a$ denotes the period of the 1D frame. The chiral polarization $\Pi = \langle x^A \rangle - \langle x^B \rangle$ is negative: the chain is left polarized regardless of the choice of unit cell. **b.** The winding number of the Bloch Hamiltonian encodes the chiral polarization *relative* to a given unit cell. The chiral polarization being a material property, the winding number $w$ can therefore take any integer value when redefining the geometry of the unit cell as illustrated in the last column. Whatever the choice of the unit cell, the difference between the geometrical polarization and $aw$ has a constant value given by the chiral polarization $\Pi$.

In Eq. (5) the $p_j$ are the components of the geometrical-polarization vector connecting the centers of mass of the $A$ and $B$ sites in the unit-cell:

$$p = \sum_{\alpha \in A} r_\alpha - \sum_{\alpha \in B} r_\alpha. \tag{7}$$

In crystals, Eqs. 5 quantifies the difference between the polarity of the ground-state wave function $\Pi$ and the geometric polarization of the frame $p$. This difference is finite only when the two sublattice Zak phases differ.

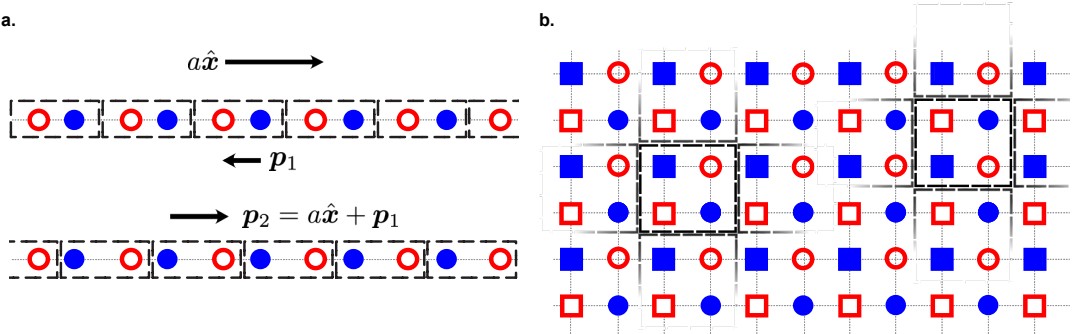

Figure 3: **Inferring the band topology from frame geometry**. **a.** The two-sites Wigner-Seitz cell on a 1D chiral frame have different geometrical polarizations; their difference is given by one Bravais vector. Consequently, we can always define the unit cell so that the Bloch Hamiltonian has a finite winding. **b.** All the Wigner-Seitz unit cells on the checkerboard lattice share the same (vanishing) chiral polarization. Therefore a single winding number $w$ characterizes the Hamiltonians on this frame in virtue of Eq. (11). Evaluating the winding using the Wigner-Seitz cell compatible with the atomic limit of $\mathcal{H}$ yields $w = 0$, by definition.

## 3 Topology of chiral insulators

We now elucidate the relation between the chiral polarization and the band topology of chiral gapped phases defined on periodic lattices. We outline the demonstrations of our central results below and detail them in Appendix B.

### 3.1 Sublattice Zak phases and winding numbers.

Computing the Wilson loop of the non-Abelian connection $\mathbf{A}_{n,m}(\boldsymbol{k}) = \langle u_n(\boldsymbol{k})| \partial_{\boldsymbol{k}} |u_m(\boldsymbol{k})\rangle$ along $\mathcal{C}_j$, we show that chirality relates the $d$ Zak phases $\gamma_j^A + \gamma_j^B$ to the windings of the Bloch Hamiltonian as

$$\gamma_j^A + \gamma_j^B = \pi w_j + 2\pi\mathbb{Z}, \tag{8}$$

where $w_j = i/(4\pi) \int_{\mathcal{C}_j} d\boldsymbol{k} \cdot \mathrm{Tr}\left[\partial_{\boldsymbol{k}} H \mathbb{C} H^{-1}\right] \in \mathbb{Z}$. The total Zak phase is quantized but the arbitrary choice of the origin of space implies that both $\gamma^A$ and $\gamma^B$ are only defined up to an integer. As a matter of fact, a mere $U(1)$ gauge transformation $|u_n\rangle \to e^{i\alpha_n(k)}|u_n\rangle$ arbitrarily modifies $\gamma_j^A(n)$ and $\gamma_j^B(n)$ by the same quantized value: $\gamma_j^A(n) \to \gamma_j^A(n) + \pi m$, $\gamma_j^B(n) \to \gamma_j^B(n) + \pi m$, with $m \in \mathbb{Z}$. By contrast, the difference between the two sublattice Zak phases is left unchanged by the same gauge transformation which echoes its independence from the space origin. Evaluating the winding of $H(\boldsymbol{k})$ using the Bloch eigenstates (see Appendix A), we readily establish the essential relation[2]

$$\gamma_j^B - \gamma_j^A = \pi w_j \quad \in \pi\mathbb{Z}. \tag{9}$$

Chirality quantizes the sublattice Zak phases of chiral insulators, even in the absence of inversion or any other specific crystal symmetry. $\gamma_j^A$ and $\gamma_j^B$ are however not independent. Combining Eqs. (8) and (9) we can always define the origin of space so that $\gamma_j^A = 0$ and $\gamma_j^B = \pi w_j$.

The $d$ winding numbers of Eq. (9) characterize the topology of $H(\boldsymbol{k})$. In particular, if for a given Wigner-Seitz cell the corresponding $H(\boldsymbol{k})$ is associated to a finite winding ($w_j \neq 0$),

---

[2]Note that this difference of Zak phases was recently denoted as a chiral phase index in [44].

then it cannot be smoothly deformed into the atomic limit defined over the same unit cell. We recall that the atomic limit of a material corresponds to a smooth deformation of the couplings to separate the energy scales so that the Wannier functions are exponentially localized, and respect the symmetries of the crystal [45]. In practice, it consists in choosing a unit cell including the strongest couplings.

The set of winding numbers is however poorly informative about the spatial distribution of the charges in electronic systems, or about the stress and displacement distributions in mechanical structures. The values of $w_j$ are defined only up to the arbitrary choice of unit cell required to construct the Bloch theory. A well known example of this limitation is given by the SSH model, where the winding of $H_k$ can either take the values 0 or $\pm 1$ depending on whether the unit cell's leftmost site belongs to the $A$ or $B$ sublattice, see Fig. 2a and Appendix B. We show in the next section, how the chiral polarization alleviates this limitation.

## 3.2 Disentangling Hamiltonian topology from frame geometry.

Equations (5) and (9) provide a clear geometrical interpretation of the winding number $w_j$ as the quantized difference between the geometrical and the chiral polarization:

$$\Pi_j = \left( p_j - a_j w_j \right). \tag{10}$$

We can now use this relation to clarify the definition of a chiral topological insulator. The chiral polarization $\Pi_j = 2\langle \mathbb{C} x_j \rangle_{E<0}$ is a physical quantity that does not depend on the specifics of the Bloch representation. Therefore computing $\Pi_j$ for two unit cells (1) and (2), we find that the windings of the two corresponding Bloch Hamiltonians $H^{(1)}(\boldsymbol{k})$ and $H^{(2)}(\boldsymbol{k})$ are related via Eq. (10) as

$$w_j^{(2)} - w_j^{(1)} = \frac{1}{a_j} \left( p_j^{(2)} - p_j^{(1)} \right). \tag{11}$$

This essential relation implies that one can always construct a Bloch representation of $\mathcal{H}$ where $H(\boldsymbol{k})$ is topologically trivial, at the expense of a suitable choice of a unit cell. As a matter of fact, a redefinition of the unit cell can increase, or reduce the geometrical polarization, and therefore the winding numbers, by an arbitrary large multiple of $a_j$ as illustrated in Fig. 2b.

For instance in the case of Hamiltonians with nearest neighbor couplings, applying Eq. (11) to Wigner Seitz unit cells ($|w_j| \leq 1$), we find that there exist as many topological classes of $\mathcal{H}$, as different geometrical polarizations in the Wigner-Seitz cells. This number provides a direct count of the chiral 'atomic limits' of $\mathcal{H}$.

Defining the topology of a chiral material therefore requires characterizing both the winding of its Bloch Hamiltonian, and the frame geometry. Remarkably, this interplay provides an insight on topological band properties from the sole inspection of the frame structure.

## 3.3 Inferring band topology from frame geometry.

There exists no trivial chiral phase in one dimension: one can always choose a Wigner-Seitz cell such that the Bloch representation of $\mathcal{H}$ has a non-vanishing winding. As a matter of fact, the geometrical polarization of the Wigner-Seitz cells can only take two finite values of opposite sign depending on whether the leftmost site in a unit cell is of the $A$ or $B$ type, see Fig. 3a. Equation (11) therefore implies that, in $1D$, there always exists, at least, two topologically distinct gapped phases smoothly connected to two atomic limits. The two gapped phases are characterized by two distinct pairs of winding numbers defined by two inequivalent choices of unit cells. In other words all SSH Hamiltonians are topological.

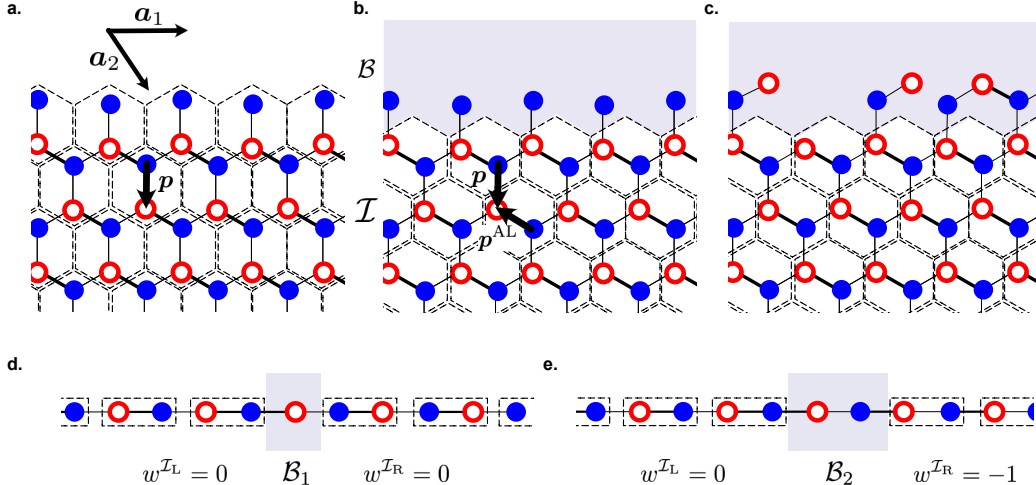

Figure 4: **Bulk-boundary correspondance. a.** A chiral crystal defined on a honeycomb frame is terminated by a clean zigzag edge incompatible with the atomic-limit Hamiltonian defined by keeping only the dominant couplings represented by thick solid lines. The dashed rectangles indicate the Wigner-Seitz cells allowing a tessellation compatible with the edge geometry. The arrow indicates the geometrical polarization $p$. **b.** Same physical system. The crystalline bulk is now tiled using the unit cell compatible with the atomic limit. This requires a redefinition of the crystal boundary $\mathcal{B}$ (shaded region). The arrows indicate the geometrical polarization of the new unit cell ($p^{\mathrm{AL}}$). The difference $p - p^{\mathrm{AL}}$ is a Bravais lattice vector ($a_2$). **c.** Same material as in (**a**.) and (**b**.) including a disordered interface $\mathcal{B}$ bearing a non-zero chiral charge $\mathcal{M}^{\mathcal{B}}$. **d.** Two connected SSH chains. The Wigner-Seitz cell in the two materials are compatible with their atomic limits. The interface $\mathcal{B}_1$ separating the two materials is one-site wide. **e.** Redefining the Wigner-Seitz cell on the right hand side of the interface requires widening the boundary region. This redefinition makes the unit cell incompatible with the atomic limit. The winding of the Bloch Hamiltonian in $\mathcal{I}_{\mathrm{R}}$ takes a finite value and consequently modifies the zero-mode content of the boundary region.

Similarly, in $d > 1$ only frames having a geometrical polarization invariant upon redefinition of the Wigner-Seitz cell can support topologically trivial Hamiltonians. Equation (11) indeed implies that a topologically trivial Hamiltonian $\mathcal{H}$ constrains the frame geometry to obey $p_j^{(1)} = p_j^{(2)}$ for all pairs of unit cells and in all directions $j$. We show a concrete example of such a frame in Fig. 3b.

Before discussing the crucial role of the frame topology and geometry on the bulk-boundary correspondence of chiral phases, we extend these two notions to chiral insulators with a flat band.

### 3.4 Chiral polarization in the presence of a net chiral charge.

It is worth noting that the chiral polarization can also be defined and computed in the presence of an additional zero-energy flat band in the gap. As detailed in the Appendix C section, it then takes the form

$$\Pi_j = (p_j - p_j^{\mathrm{ZM}}) + a\left(\gamma_j^A - \gamma_j^B\right)/\pi\,. \tag{12}$$

In this case, we loose the clear decomposition $\Pi$ into geometrical and topological contributions. The geometrical polarization is corrected by $\boldsymbol{p}^{\text{ZM}}$ which originates from a spectral contribution associated to the zero-energy band. Furthermore the second term on the r.h.s., the difference between two geometrical Zak phases, is not a topological winding number anymore. Despite the seemingly complex form of Eq. (12), we show in the next section that the chiral polarization remains an effective tool to relate spectral bulk properties to the number of zero-energy states localized at boundaries.

# 4 Bulk-boundary correspondence

We now establish a bulk-boundary correspondence relating the chiral polarization to the number of zero modes supported by the free surface of a chiral insulator. For the sake of clarity, we discuss the two-dimensional case without loss of generality. We consider first a crystalline insulator $\mathcal{I}$ terminated by a clean edge $\partial\mathcal{I}$ oriented along a Bravais vector, say $\boldsymbol{a}_1$ as illustrated in Fig. 4a.

The bulk of the insulator can be described by different types of unit cells. As illustrated in Fig. 4a, in the presence of a clean edge, it is natural to choose a unit cell which allows a tessellation of the whole system. However, this unit cell is generically incompatible with the atomic limit of the Hamiltonian, and therefore does not allow a direct count of the zero energy boundary states using the simple Maxwell-Calladine count. An obvious strategy hence consist in redefining the unit cell, as in Fig. 4b to match the constraints of the atomic limit. This redefinition comes at the expense of leaving sites outside of the bulk tessellation. We define this ensemble of sites as the boundary region $\mathcal{B}$. Keeping in mind that we can smoothly deform the Hamiltonian into its atomic limit without closing the gap, we use Eq. (2) to count the number of zero energy states hosted by $\mathcal{B}$. It is given by $\mathcal{V} = \mathcal{M}^{\mathcal{B}}$. An essential geometrical observation is that the net chiral charge in $\mathcal{B}$ can be expressed as $\mathcal{N}^{\partial\mathcal{I}}(p_2^{\text{AL}} - p_2)$, where $\mathcal{N}^{\partial\mathcal{I}}$ is the edge length expressed in number of unit cells and $p_2$ is the geometrical polarization of the initial unit cell. We can now make use of the invariance of the chiral polarization formalized by Eq. (11) to relate the geometrical count of zeromodes to the winding of the Bloch Hamiltonian: $\mathcal{V} = \mathcal{N}^{\partial\mathcal{I}}(p_2^{\text{AL}} - p_2) = \mathcal{N}^{\partial\mathcal{I}} w_2^{\mathcal{I}}$. To arrive at a bulk boundary correspondence generic to all chiral insulators, we include the possibility of dealing with irregular interfaces featuring a net chiral charge $\mathcal{M}^{\mathcal{B}}$ as sketched in Fig. 4c. We then find

$$\mathcal{V} = \mathcal{M}^{\mathcal{B}} + \mathcal{N}^{\partial\mathcal{I}} w_2^{\mathcal{I}}. \tag{13}$$

Three comments are in order. Firstly, the bulk boundary correspondence defined by Eq. (13) reveals the geometrical implication of a nonzero winding: a finite $w_j^{\mathcal{I}}$ echoes the impossibility to tile a periodic frame with unit cells compatible with the Hamiltonian's atomic limit. Secondly, Eq. (13) is readily generalized to interfaces separating two chiral insulators $\mathcal{I}_{\text{L}}$ and $\mathcal{I}_{\text{R}}$, where we simply have to apply the same reasoning on each side of the interface: $\mathcal{V} = \mathcal{M}^{\mathcal{B}} + \mathcal{N}^{\partial\mathcal{I}}(w^{\mathcal{I}_{\text{L}}} + w^{\mathcal{I}_{\text{R}}})$, see e.g. Figs. 4d and 4e. Thirdly, the formula given by Eq. (13) generalizes the Kane-Lubensky index introduced in their seminal work to count the zero-energy modes localized within isostatic mechanical networks [25]. We show that this index defines a bulk-boundary correspondence generic to all chiral insulators and even to flat band insulators such as hyperstatic lattices as further discussed in the Appendix C.

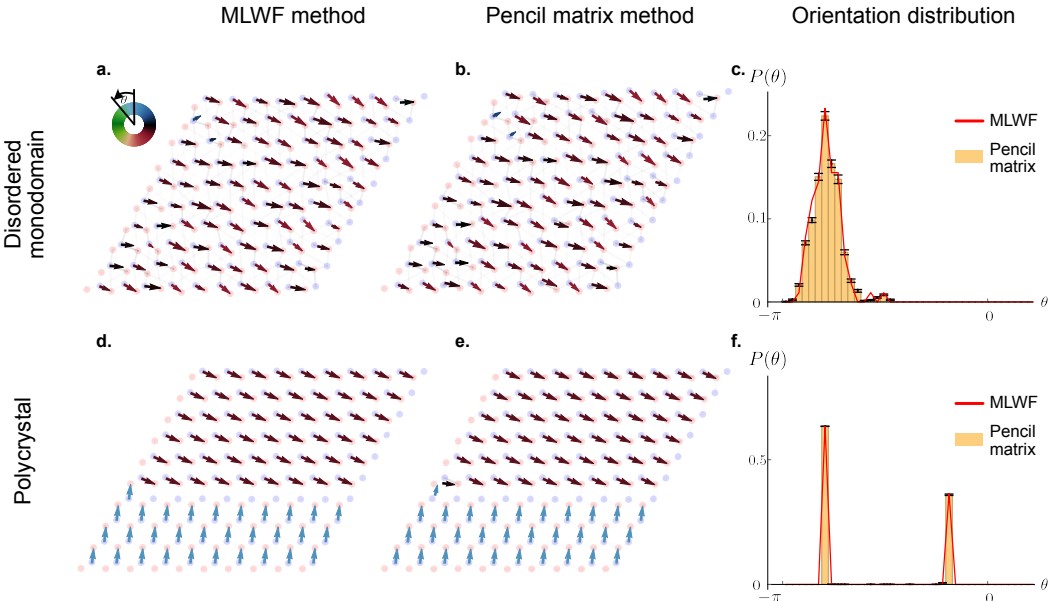

Figure 5: **Pencil matrix versus maximally localized Wannier functions a.** Single domain configuration with geometrical and spectral disorder. The chiral polarization field obtained from the maximally localized wannier functions is superposed. **b.** Chiral polarization field obtained from one realization of the pencil matrix procedure. **c.** Orientation distribution obtained from 50 values of $\alpha$ (bar chart), and from the maximally localized wannier functions (red solid line). **d.**, **e.**, **f.** correspond to the same information, this time for two crystalline domains.

# 5 Amorphous Chiral Insulators

In condensed matter, chiral symmetry is a low energy feature of electronic Hamiltonians, which is unlikely to survive to strong structural disorder. Conversely, in photonic, accoustic or mechanical metamaterials chirality is built in by design and can therefore be present both in ordered or amorphous structures [20, 46]. In mechanical metamaterials chirality is even more robust as it is inherent to any system assembled from elastically coupled degrees of freedoms [21]. In this section, we show how to generalize our physical characterization of zero energy modes to disordered chiral metamaterials.

Over the past two years a number of experimental, numerical and theoretical works showed that crystalline symmetries are not required to define topological insulators, see e.g. [47–50]. Unlike these pionneering studies where topologically inequivalent disordered phases are distinguished by abstract indices defined in real space and related to the quantification of edge currents, our framework solely based on the chiral polarization applies to chiral systems regardless of the presence or not of time reversal symmetry.

Our strategy follows from the fundamental relation: $\Pi_j = p_j - a_j w_j$ of Eq. (10). This relation implies a one-to-one correspondence between the chiral polarization and a topological spectral property quantized by the winding vector. The basic idea hence consists in probing the existence of topologically protected zero modes by *local* discontinuities in the chiral polarization field, even when no winding number or Zak phase can be defined. Relating topologically protected excitations to real-space singularities requires defining a local chiral polarization field $\Pi(x)$. By definition, $\Pi(x)$ measures the local imbalance of the wave function carried by

the $A$ and $B$ sites. To express $\mathbf{\Pi}(\boldsymbol{x})$, it would be natural to consider eigenstates of the position operator $P\boldsymbol{x}P$ projected onto the occupied states of $\mathcal{H}$. However, in dimension $d > 1$, the different components of the projected position operator do not commute $[Px_jP, Px_kP] \neq 0$ for $j \neq k$, and do not possess common eigenstates. Instead, we express the polarisation in terms of the maximally localized states $\widetilde{W}_m$ [28], which are centered on the position $\boldsymbol{x}_m \equiv \langle \widetilde{W}_m | \hat{X} | \widetilde{W}_m \rangle$. These states generalize the Wannier functions in the absence of translational symmetry, see Appendix E for more details. We can then define the *local* chiral polarization as the weighted chiral position evaluated over $\widetilde{W}_m$:

$$\mathbf{\Pi}(\boldsymbol{x}_m) = 2 \langle \widetilde{W}_m | \mathbb{C}\hat{X} | \widetilde{W}_m \rangle . \tag{14}$$

In practice, we can bypass the time consuming numerical determination of the $\widetilde{W}_m$ by taking advantage of the matrix pencil method detailed in Appendix D. In short, the method consists in replacing in (14) the $\widetilde{W}_m$ by eigenstates of a linear combination of the projected position components $L = \sum_j \alpha_j Px_jP$ ; $\sum \alpha_j = 1$. The dependence on $\alpha_i$ of the resulting chiral polarization is a measure of the non-commutativity of the $Px_j$ typically associated to a nonvanishing Berry curvature. In practice, as illustrated in Fig. 5, the difference between the actual polarization, computed from the $\widetilde{W}_m$, and its approximation based on the $R$-matrix eigenstates is smaller than the distance between neighboring sites. Given the excellent agreement found both in mono and polycrystals, we henceforth use the pencil matrix method to locally measure the chiral polarization fields in disordered and amorphous structures out of reach of conventional chiral displacement characterizations [51].

To make the discussion as clear as possible we consider separately the two possible sources of randomness in a disordered chiral insulator: (i) geometrical disorder, which affects the frame geometry leaving the interaction between the $A$ and $B$ sites unchanged and (ii) Spectral disorder, which alters the interactions while living the frame geometry unchanged.

### 5.1 Topological zero modes on amorphous chiral frames.

The reasoning is easily explained starting from a concrete example. Fig. 6 shows the interface between two topologically distinct insulators, $\mathcal{I}_\mathrm{T}$ and $\mathcal{I}_\mathrm{B}$, living on a honeycomb frame. They correspond to distinct atomic limits of a nearest-neighbor tight binding Hamiltonians including two different hopping coefficients, see e.g. [52]. For the choice of unit cell sketched in Fig. 6, the winding vectors are $\boldsymbol{w}^{\mathcal{I}_\mathrm{T}} = (0, 1)$ and $\boldsymbol{w}^{\mathcal{I}_\mathrm{B}} = (1, 0)$. As a result the boundary region $\mathcal{B}$ hosts one zero mode per unit cell located on the $A$ sites. As expected from Eq. (10), on a homogeneous periodic frame, $\mathbf{\Pi}(\boldsymbol{x})$ takes two distinct values in the two regions, and is discontinuous across $\mathcal{B}$. Correspondingly, the distribution of the chiral polarization in the sample consists of two peaks centered on the two values associated to two topologically inequivalent phases, see Fig. 6 (left column).

We now disorder the frame by shifting all site positions by random displacements of maximal amplitude $|\delta\boldsymbol{x}|$ while preserving the magnitude of the interactions in the corresponding Hamiltonian $\mathcal{H}_\mathrm{D}$. For sufficiently large displacements, it is impossible to keep track of the original periodic lattice, see Fig. 6 (first row). Nonetheless, we clearly see in the third row of Fig. 6 that the topologically protected zero modes located in $\mathcal{B}$ are preserved, despite the lack of crystalline symmetry and the impossibility to define a Bloch Hamiltonian and its topological winding numbers. Note that unlike in [53] both the bulk and the boundary region are homogeneously disordered. Again, the existence and location of a line of zero modes is revealed by variations of the chiral polarization field. The variations of the orientation of $\mathbf{\Pi}(\boldsymbol{x})$ occurs over the penetration length-scale $\ell_\mathrm{G}$ set by the energy gap. The coexistence of two topologically distinct amorphous phases is signalled by a (wider) bimodal distribution of $\Pi(\boldsymbol{x})$ peaked

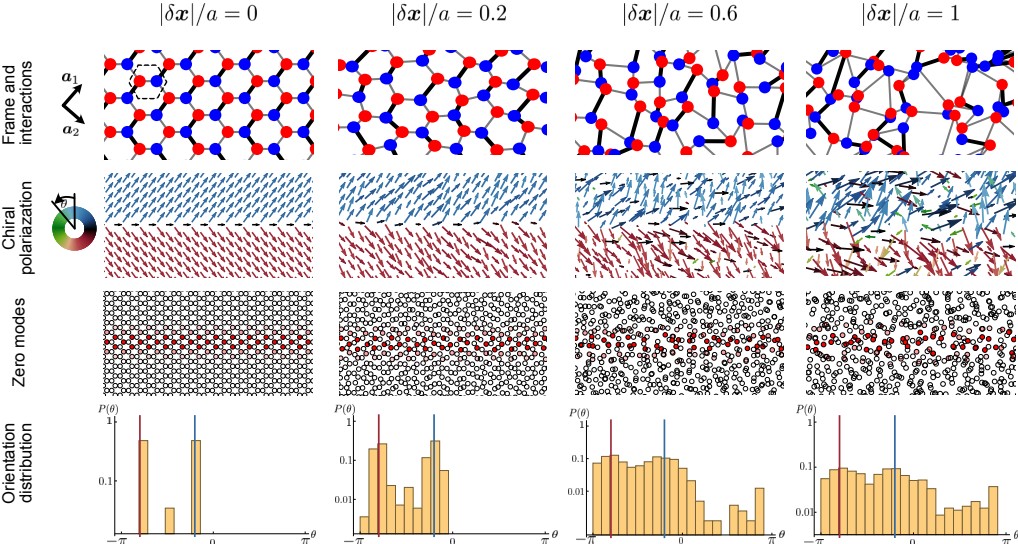

Figure 6: **Topological zero energy states on amorphous frames** First row: Sketch of the frame geometry for increasing positional disorder quantified by the maximal amplitude of the random displacements $|\delta \boldsymbol{x}|/a$. All panels show the vicinity of a boundary between two different insulators defined on the same frame but with different positions of the stronger couplings. The lines' width indicates the magnitude of the coupling strength. In all panels $t'/t = 20$. In the leftmost panel, we indicate the choice of the unit cell and of the crystallographic axes. Second row: Corresponding chiral polarization fields. The color indicates the orientation of $\boldsymbol{\Pi}(\boldsymbol{x})$ Third row: Magnitude of the zero-mode wave function. The zero mode is located at the boundary between topologically inequivalent states even on amorphous frames. Fourth row: Probability density function of the $\theta$, the local orientation of the chiral polarization field. The distributions are peaked on the same two directions (vertical lines) regardless of the magnitude of disorder. This reveals the coexistence of two distinct topological phases robust to positional disorder.

on the same values as in the pure case, see Fig. 6 (last row). This robust phenomenology is further illustrated in Supplementary Video 1, showing the evolution of the polarization field and zero-mode location as the magnitude of disorder is increased.

This observation reflects a generic feature of chiral matter. Randomizing the frame geometry cannot alter the energy gap provided that the graph defined by the coupling terms of $\mathcal{H}$ has a fixed chiral connectivity. This observation implies that the concept of topological phase naturally applies to amorphous frames that can be continuously deformed into periodic lattices. In fact, the coexistence of different chiral insulators is effectively probed by the spatial distribution of the polarization field $\Pi(\boldsymbol{x})$. Each peak of the distribution signals topologically inequivalent regions in amorphous chiral matter. The phase boundaries are then readily detected by jumps of the chiral-polarization vector field over $\ell_G$.

## 5.2 Topological zero modes of disordered chiral Hamiltonians.

The case of spectral disorder is more subtle as it can trigger topological transitions. Again, we start with a concrete example. We use the same model of insulator as in the previous section. Considering the even simpler case of a perfect monocrystal, there is no zero mode



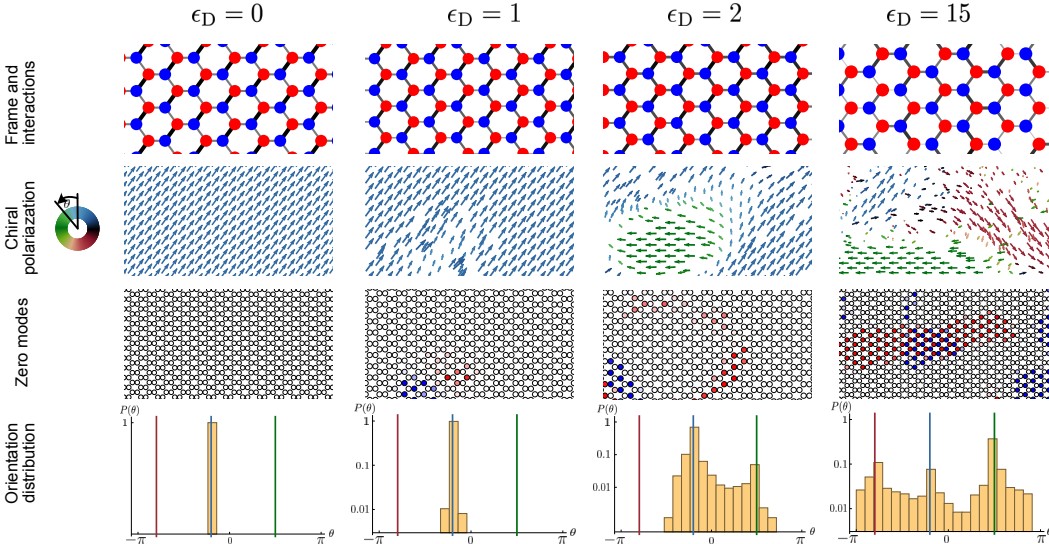

Figure 7: **Topological zero-energy states in the bulk of disordered chiral insulators** First row Sketch of the honeycomb frame and of the coupling strengths for increasing spectral disorder. The strengths of the couplings are represented by the width of the dark lines. Their randomness is quantified by the variance of the Gaussian couplings $\epsilon_D$. The correlation length for all the examples is $\xi = 12a$. Second row Corresponding chiral polarization fields. The color indicates the orientation $\theta$ of $\mathbf{\Pi}(\boldsymbol{x})$. Third Row: Magnitude of the zero-energy modes on the $A$ (red) and $B$ (blue) sites. Fourth row: Probability density function of the orientation $\theta$. Remarkably, even in the disordered cases, the distribution peaks only at values characteristic of the three phases of the homogeneous chiral Hamiltonian.

in the sample. Keeping the frame unchanged we add disorder to the interactions in the form of random perturbations to the coupling parameters. We note $\epsilon_D$ the width of the Gaussian disorder distribution, $\xi$ its correlation length and $\Delta E$ the energy gap in the pure case. When $\epsilon_D/\Delta E- \ll 1$ no zero mode exists in this finite system see Fig. 7 first column. Consistently, the local chiral polarization hardly fluctuates in space and its distribution remains peaked around the same constant value.

By contrast as $\epsilon_D/\Delta E \sim 1$, zero energy modes emerge in the bulk. Their presence signals local the emergence of topologically inequivalent regions in the material triggered by local gap inversions. The distinct phases are revealed by the orientational distribution of $\mathbf{\Pi}(\boldsymbol{x})$: as disorder increases additional peaks grow at values of $\theta$ characteristic of the other two homogeneous topological insulators, Fig. 7 (last row). In the limit of strong disorder, the spatial extent of the coexisting phases is set by the disorder correlation length $\xi$ as exemplified in Supplementary Movie 2. Gap closings also have a local signature in the polarization field. As $\mathbf{\Pi}(\boldsymbol{x}_m)$ is only defined at the generalized Wannier centers (Eq. (14)), $\mathbf{\Pi}(\boldsymbol{x}_m)$ cannot be computed at the center of a zero mode, which by definition does not support any Wannier mode. The proliferation of zero modes in the bulk is therefore signaled by an increasing number of holes in the polarization field.

The above observations do not rely on the specific model we use in Figs. 6 and 7. Generically, adding spectral disorder to a chiral Hamiltonian results in the nucleation of additional topological phases decorated by zero modes at their boundaries. Even in the absence of a Bloch theory, we can distinguish the topological nature of the coexisting phases by measuring

their average chiral polarization. For spatially correlated disorder the spatial extent of each phase is set by the disorder correlation length $\xi$.

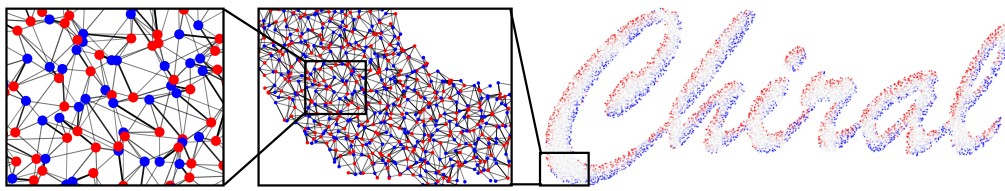

Figure 8: **Disordered chiral metamaterial** Macroscopic view and close ups on an amorphous frame supporting a disordered chiral insulator. The frame is defined adding a strong positional disorder to a Honeycomb lattice $|\delta x| = a$. Using the same Hamiltonian as in Figs.6 and 7, we add spectral disorder corresponds to $\epsilon_D = 2$. Cutting the sample to form the word "chiral" reveals a continuous distribution of zero modes along the edge.

## 5.3 Designing topologically protected zero modes in amorphous chiral matter.

It is worth stressing that disordered chiral insulators generically support topologically protected zero-energy modes at their boundaries. Unlike crystaline topological insulators, the lower the bulk and edge symmetries the more robust the edge states.

Cutting an amorphous sample into two parts without inducing the proliferation of boundary zero modes is virtually impossible. It would require cutting bonds while preserving the connectivity between all pairs of $A$ and $B$ site connected by the local polarization vectors $\mathbf{\Pi}(x_m)$; only this type of configurations can be continuously deformed into crystals having edges matching that of tilings generated by the unit cell of an atomic limit. These cuts require extreme fine tuning in macroscopic samples and are therefore virtually impossible to achieve. This property makes the design of zero energy wave guides very robust in amorphous chiral matter. As illustrated in Fig. 8.

## 5.4 Measuring the chiral polarization.

In this section we show that the chiral polarization is not only a powerful theoretical concept, but an actual material property readily accessible to experiments. Two scenarios are possible: when the (low energy) eigenfunctions can be measured, the chiral polarization can be directly evaluated using its definition, Eq. (3). This technique is straightforward e.g. in mechanical metamaterials [54], where the vibrational eigenmodes can be imaged in real space in response to mechanical actuation.

Alternatively, when spectral properties are out of reach of quantitative measurements, we can infer the value of the chiral polarization from the dynamic spreading of localized chiral excitations. This approach builds and generalizes the technique pioneered in the context of periodically driven photonic quantum walk [55, 56]. For the sake of clarity we henceforth limit our discussion to 1D, two-band insulators although the reasonning applies in higher dimensions.

We introduce the dynamical chiral polarization $\Pi_\Psi(t) = \langle \Psi(t)|\mathbb{C}\hat{X}|\Psi(t)\rangle$ defined over the time-evolved states $\Psi(t) = \exp(-iHt)\Psi(0)$, where $\Psi(0)$ is a localized chiral state. Should one be able to initalize an experiment in a Wannier State $\Psi(0) = W_{n,\boldsymbol{R}}$, the wave function would spread as in Fig. 9a, but remarkably the dynamical chiral polarization $\Pi_\Psi(t)$ would be stationnary and equal to $\Pi$ in a homogeneous system as illustrated in Fig. 9a, and demonstrated in the Method section. In practice, it would be always easier to approximate the Wannier state by excitations $\Psi_{AB}$ (resp. $\Psi_{BA}$) localized on two neighboring $A$ and $B$ sites (resp. $B$ and $A$). The result of this procedure is shown in Fig. 9b and reveals that the long-time dynamics of $\Pi_\Psi(t)$ converges towards the chiral polarization $\Pi$. However, we stress that the essential information about the orientation of $\Pi$ is already accessible at very short times and would not suffer from possible damping issues. When $\Pi_\Psi(t=0)$ and $\Pi$ have opposite signs, we observe very large amplitude oscillations reflecting the dynamic reversal of the chirality of the wave packet at short times. Conversely when $\Pi_\Psi(t=0)$ and $\Pi$ are parallel the convergence is very fast and devoid of large amplitude fluctuations.

It is worth noting that the chiral initial state $\Psi(t=0) = \Psi_{AB}$ is an atomic-limit eigenstate. The dynamics can then be seen as the result of a quench at $t=0$ starting from the atomic-limit Hamiltonian. The amplitude of the fluctuations in Fig. 9b then reveals the topological nature of the quench. As a last comment we stress that our protocol differs from the chiral displacement method introduced and used in [55–58]. The mean chiral displacement depends on the unit-cell convention. As a consequence, to probe the topology of 1D systems, conventional MCD protocols require two independent measurement protocols. They effectively correspond to measuring the mean chiral displacement given two possible unit cell choices. A topological invariant is then defined by the difference between the two measurements. The Chiral polarization method, which we introduce provides a one-step characterization of the topology of a chiral phase.

## 6 Conclusion

We have established a generic framework to characterize, elucidate and design the topological phases of chiral insulators. In crystals, we show that the frame topology and the frame geometryact together with Bloch Hamiltonian topology to determine the zero-mode content of the bulk and interfaces. In the bulk, the frame topology fully determines the algebraic number of zero-energy modes counted by the chiral charge $\mathcal{M}$. Chiral insulators, however, are distinguished one another via their chiral polarization $\boldsymbol{\Pi}$ set both by the frame geometry and Bloch-Hamiltonian topology. At their surface, the number of zero-energy states is prescribed by the interplay between the Bloch Hamiltonian topology and the frame geometry in the bulk on one hand, and by the frame topology of the boundary on the other hand. This interplay goes beyond the bulk-boundary-correspondence principles solely based on Hamiltonian topology.

We have shown that chiral symmetry alone translates real-space properties into spectral phases without relying on any crystalline symmetry and translational invariance when expressed as a sublattice symmetry. Chiral symmetry does not merely complement the classification of topological quantum chemistry [45, 59–61] but also makes it possible to distinguish topological phases in amorphous matter. In disordered system, introducing the concept of chiral polarization field, we provide a practical platform to detect topological phases coexisting in disordered samples, an to design robust zero-mode wave guides at their boundaries.

We expect our framework to extend beyond Hamiltonian dynamics when dissipative processes obey the chiral symmetry [62]. We therefore conjecture that real-space topology, geometry and non-Hermitian operator topology should cooperate in chiral dissipative materials as diverse as cold atoms to photonics, robotic devices and active matter.

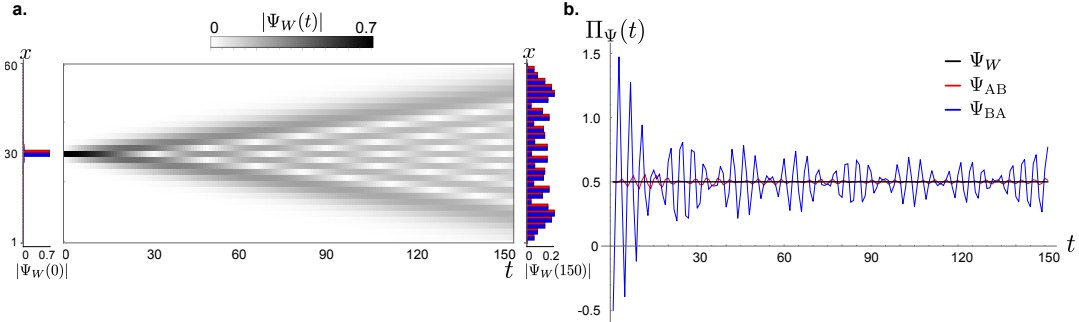

Figure 9: **Measuring the chiral polarization in time. a.** Left: Dynamical evolution of a Wannier state in the ground state of a two-band SSH model, with hopping ratio $t_1/t_2 = 0.1$. The state is localized in the middle of a finite system of 60 unit cells. Center: Time evolution of the wave-function amplitude. Right: The amplitude of the final state at time $t = 250$ is represented on the A (red) and B (blue) sites of the lattice. **b.** The dynamical chiral polarization $\Pi_\psi(t) = \langle \Psi(t)|\mathbb{C}\hat{X}|\Psi(t)\rangle$ corresponding to the protocol described in **a** is constant in time (black solid line). By comparison, the dynamical chiral polarization starting from a state $\Psi_{AB}(t=0)$ (resp. $\Psi_{BA}(t=0)$), localized on two neighboring sites $A$ and $B$ (resp. $B$ and $A$) shows fluctuations around the static chiral polarization whose amplitude depends on the initial state. The sign $\Pi_\psi(t)$ is reversed at short time when the chiral polarization of the initial state is opposite to the static chiral polarization of the SSH chain. This results in large amplitude oscillations. The short time dynamics of $\Pi_\psi$ therefore provides a direct access to the orientation of the material chiral polarization.

## Acknowledgements

We thank J. Asboth, A. Bernevig, A. Dauphin, K. Gawedzki, A. Grushin, Y. Hatsugai, P. Massignan, A. Po, A. Schnyder and A. Vishwanath for insightful discussions.

**Funding information** We acknowledge support from ANR WTF, and ToRe IdexLyon breakthrough programs.

## A Bloch theory convention and Wannier states.

### A.1 Conventions for the Bloch decomposition.

For the sake of clarity, we first introduce the main quantities used throughout all the manuscript to describe waves in periodic lattices. We note $\left|\Psi_{n,\boldsymbol{k}}\right\rangle$ the Bloch eigenstates. They correspond to wavefunctions $\left\langle \boldsymbol{x}|\Psi_{n,\boldsymbol{k}}\right\rangle = \varphi_{n,\boldsymbol{k}}(\boldsymbol{x})e^{i\boldsymbol{k}\cdot\boldsymbol{x}}$, where $\boldsymbol{k}$ is the momentum in the Brillouin Zone (BZ), and where the normalized function $\varphi_{n,\boldsymbol{k}}$ has a periodicity of one unit cell [28]. In this article, we use the following convention to express the Bloch states as a superposition of plane waves:

$$\left|\Psi_{n,\boldsymbol{k}}\right\rangle = \sum_\alpha u_{n,\alpha}(\boldsymbol{k})\left|\boldsymbol{k},\alpha\right\rangle, \tag{15}$$

where $\alpha$ labels the different atoms in the crystal, and $|\boldsymbol{k},\alpha\rangle$ represents the Fourier transform of the real-space position basis: $|\boldsymbol{k},\alpha\rangle = \sum_{\boldsymbol{R}}\exp(i\boldsymbol{k}\cdot\boldsymbol{R})|\boldsymbol{R}+\boldsymbol{r}_\alpha\rangle$, $\boldsymbol{R}$ being a Bravais lattice vector and $\boldsymbol{r}_\alpha$ a site position within the unit cell. We stress that here the components $u_{n,\alpha}(\boldsymbol{k})$

are periodic functions of $\boldsymbol{k}$ over the BZ. It is worth noting, however, that there exists multiple conventions to decompose the Bloch states as discussed e.g in the context of graphene-like systems in [63–65]. A common alternative uses nonperiodic components over the BZ which carry an additional phase encoding the position of each atom within the unit cell: $\left|\Psi_{n,\boldsymbol{k}}\right\rangle = \sum_{\alpha} \tilde{u}_{n,\boldsymbol{k},\alpha} e^{i\boldsymbol{k}\cdot\boldsymbol{r}_{\alpha}} \left|\boldsymbol{k},\alpha\right\rangle$. We will comment on the translation of our results from one convention to the other in the following.

## A.2 Wannier functions.

By definition the Wannier function associated to a Bloch eigenstate is given by the inverse Fourier transform (up to a phase):

$$\left|W_{n,\boldsymbol{R}}\right\rangle = \Omega^{-1} \int_{\text{BZ}} \mathrm{d}^d \boldsymbol{k} \, e^{-i\boldsymbol{k}\cdot\boldsymbol{R}} \left|\Psi_{n,\boldsymbol{k}}\right\rangle . \tag{16}$$

where $\Omega$ is the volume of the BZ. Note that for sake of clarity, we here and henceforth assume that the spectrum does not include band crossings. The technical generalization of our results to degenerated spectra is straightforward but involves some rather heavy algebra, see e.g. [28]. In addition, to ease the notation and calculations we work with orthogonal coordinates such that $\int_{\text{BZ}} dk_j = \Omega^{1/d}$, $\forall j$. The generalization to non-orthogonal lattices is straightforward and amounts to considering different geometrical factors along each reciprocal direction

## A.3 Projected position operator and sublattice Zak phases.

Ignoring the distinction between the $A$ and $B$ sites, we can first compute the action of the position operator on the Wannier states following [28]:

$$\begin{aligned}
\left\langle \boldsymbol{x}\right| \widehat{X} \left|W_{n,\boldsymbol{R}}\right\rangle &= \Omega^{-1} \int_{\text{BZ}} \mathrm{d}^d \boldsymbol{k} \, \boldsymbol{x} \, e^{i\boldsymbol{k}\cdot(\boldsymbol{x}-\boldsymbol{R})} \varphi_{n,\boldsymbol{k}}(\boldsymbol{x}) \\
&= \Omega^{-1} \int_{\text{BZ}} \mathrm{d}^d \boldsymbol{k} \left(-i\partial_{\boldsymbol{k}} e^{i\boldsymbol{k}\cdot(\boldsymbol{x}-\boldsymbol{R})} + \boldsymbol{R} e^{i\boldsymbol{k}\cdot(\boldsymbol{x}-\boldsymbol{R})}\right) \varphi_{n,\boldsymbol{k}}(\boldsymbol{x}) \\
&= \Omega^{-1} \int_{\text{BZ}} \mathrm{d}^d \boldsymbol{k} \, e^{-i\boldsymbol{k}\cdot\boldsymbol{R}} \left[e^{i\boldsymbol{k}\cdot\boldsymbol{x}} \left(\boldsymbol{R} + i\partial_{\boldsymbol{k}}\right)\right] \varphi_{n,\boldsymbol{k}}(\boldsymbol{x}),
\end{aligned} \tag{17}$$

where in the last step we applied an integration by parts, using that $\left|\Psi_{n,\boldsymbol{k}}\right\rangle = \left|\Psi_{n,\boldsymbol{k}+\boldsymbol{G}}\right\rangle$ with $\boldsymbol{G}$ a primitive reciprocal vector. The generalization of Eq. (17) to the position operator projected on the sublattice $a = A, B$ is straightforward:

$$\left\langle \boldsymbol{x}\right| \widehat{X}\mathbb{P}^a \left|W_{n,\boldsymbol{R}}\right\rangle = \Omega^{-1} \int_{\text{BZ}} \mathrm{d}^d \boldsymbol{k} \, e^{-i\boldsymbol{k}\cdot\boldsymbol{R}} \left[e^{i\boldsymbol{k}\cdot\boldsymbol{x}} \left(\boldsymbol{R} + i\partial_{\boldsymbol{k}}\right)\right] \mathbb{P}^a \varphi_{n,\boldsymbol{k}}(\boldsymbol{x}), \tag{18}$$

which allows us to define the average positions $\left\langle \boldsymbol{x}^a\right\rangle_{n,\boldsymbol{R}}$ restricted to the site $a = A, B$ and to the $n^{\text{th}}$ band excitations:

$$\begin{aligned}
\left\langle \boldsymbol{x}^a\right\rangle_{n,\boldsymbol{R}} &\equiv \left\langle W_{n,\boldsymbol{R}}\right| \mathbb{P}^a \hat{X} \mathbb{P}^a \left|W_{n,\boldsymbol{R}}\right\rangle \\
&= \frac{\boldsymbol{R}}{\Omega} \int_{\text{BZ}} \mathrm{d}^d \boldsymbol{k} \left\langle \varphi_{n,\boldsymbol{k}}\right| \mathbb{P}^a \left|\varphi_{n,\boldsymbol{k}}\right\rangle + \frac{1}{\Omega} \boldsymbol{\Gamma}_{\text{Zak}}^a(n),
\end{aligned} \tag{19}$$

where $\left|\varphi_{n,\boldsymbol{k}}\right\rangle = e^{-i\boldsymbol{k}\cdot\hat{X}} \left|\Psi_{n,\boldsymbol{k}}\right\rangle$, and $\boldsymbol{\Gamma}^a(n)$ is the vector composed of the $d$ generalized sublattice Zak phases associated to the $n$-th band:

$$\Gamma_j^a(n) = i \int_{\text{BZ}} \mathrm{d}^d \boldsymbol{k} \left\langle \varphi_{n,\boldsymbol{k}}\right| \mathbb{P}^a \partial_{k_j} \mathbb{P}^a \left|\varphi_{n,\boldsymbol{k}}\right\rangle . \tag{20}$$

We can further simplify Eq. (19) noting that the orthonormality of the $\left|\varphi_{n,\boldsymbol{k}}\right\rangle$ implies $\left\langle\varphi_{n,\boldsymbol{k}}\left|\mathbb{P}^A+\mathbb{P}^B\right|\varphi_{n,\boldsymbol{k}}\right\rangle=1$ and $\left\langle\varphi_{n,\boldsymbol{k}}\left|\mathbb{P}^A-\mathbb{P}^B\right|\varphi_{n,\boldsymbol{k}}\right\rangle=0$, which yields $\left\langle\varphi_{n,\boldsymbol{k}}\left|\mathbb{P}^a\right|\varphi_{n,\boldsymbol{k}}\right\rangle=1/2$. All in all, we find a simple relation between the average of the position operator and the Zak phase of the Bloch eigenstates over the BZ:

$$\langle \boldsymbol{x}^a\rangle_{n,\boldsymbol{R}} = \frac{\boldsymbol{R}}{2} + \frac{1}{\Omega}\boldsymbol{\Gamma}^a(n). \tag{21}$$

## B  Chiral polarization, Zak phases and winding.

### B.1  Chiral polarization and sublattice Zak phases.

We are now equipped to compute the chiral polarization, defined as the difference between the expected value of the projected position operators over the occupied eigenstates ($n<0$). It readily follows from Eq. (21) that $\boldsymbol{\Pi}$ corresponds to the difference of the sublattice Zak phases:

$$\begin{aligned}
\boldsymbol{\Pi} &\equiv 2\sum_{n<0}\left\langle\boldsymbol{x}^A\right\rangle_{n,\boldsymbol{R}}-\left\langle\boldsymbol{x}^B\right\rangle_{n,\boldsymbol{R}}\\
&=\frac{2}{\Omega}\sum_{n<0}\boldsymbol{\Gamma}^A(n)-\boldsymbol{\Gamma}^B(n).
\end{aligned} \tag{22}$$

Two comments are in order. Firstly, the sum could have been also taken over the unoccupied states ($n>0$). As $\mathbb{C}^2=\mathbb{I}$, the sublattice phase picked up by $\left|\varphi_{n,\boldsymbol{k}}\right\rangle$ is indeed the same as that of its chiral partner $\left|\varphi_{-n,\boldsymbol{k}}\right\rangle=\mathbb{C}\left|\varphi_{n,\boldsymbol{k}}\right\rangle$. Secondly, we stress that Eq. (22) does not depend on the specific convention of the Bloch representation. This relation, however does not disentangle the respective contributions of the frame geometry and of the Hamiltonian on the chiral polarization. To single out the two contributions, we now use the specific Bloch representation (15). Given this choice, the sublattice Zak phase is naturally divided into two contributions leading to

$$\boldsymbol{\Gamma}^a(n) = \int_{\text{BZ}} \mathrm{d}^d\boldsymbol{k}\ \sum_{\alpha\in a}\left(u_{n,\alpha}^* u_{n,\alpha}\boldsymbol{r}_\alpha + i u_{n,\alpha}^*\partial_{\boldsymbol{k}}u_{n,\alpha}\right). \tag{23}$$

The first term on the r.h.s. is the intracellular contribution to the Zak phase while the second is proportional to the sublattice intercellular Zak phase following to the definitions of [43]

$$\gamma_j^a(n) \equiv i\int_{\mathcal{C}_j}\mathrm{d}\boldsymbol{k}\sum_{\alpha\in a}u_{n,\alpha}^*(\boldsymbol{k})\partial_{\boldsymbol{k}}u_{n,\alpha}(\boldsymbol{k}). \tag{24}$$

Summing Eq.(23) over all occupied bands, and using the orthogonality of the chiral component $u_{n,\alpha}$ we then recover our central result:

$$\boldsymbol{\Pi} = \boldsymbol{p} + \frac{2}{\Omega^{1/d}}\left(\gamma^A-\gamma^B\right), \tag{25}$$

where $\boldsymbol{p}=\sum_{\alpha\in A}\boldsymbol{r}_\alpha-\sum_{\alpha\in B}\boldsymbol{r}_\alpha$ is the geometrical polarization of the corresponding unit-cell and $\gamma^a=\sum_{n<0}\gamma^a(n)$. The chiral polarization is the sum of one contribution coming only from the frame geometry and one contribution characterizating the geometrical phase of the Bloch eigenstates.

## B.2 Chiral polarization in different Bloch conventions.

Although the physical content of the chiral polarization does not depend on the choice of the Bloch convention, it is worth explaining how to derive its functional form for the other usual representation where $\left|\Psi_{n,\boldsymbol{k}}\right\rangle = \sum_{\alpha} \tilde{u}_{n,\alpha}(\boldsymbol{k})e^{i\boldsymbol{k}\cdot\boldsymbol{r}_{\alpha}}\left|\boldsymbol{k},\alpha\right\rangle$. Within this convention the vector of Zak phases take the form

$$\boldsymbol{\Gamma}^{a}(n) = i \int_{\text{BZ}} \mathrm{d}^{d}\boldsymbol{k} \sum_{\alpha \in a} \tilde{u}_{n,\alpha}^{*} \partial_{\boldsymbol{k}} \tilde{u}_{n,\alpha}, \tag{26}$$

which does not allow the distinction between the geometrical and the Hamiltonian contributions to $\Pi$ when performing the sum over the occupied band in Eq. (22). This observation further justifies our choice for the Bloch representation.

## B.3 Quantization of the intercellular Zak-phase in chiral insulators.

To demonstrate the quantization of $\gamma_{j} = \gamma_{j}^{A} + \gamma_{j}^{B}$, we resort to the Wilson loop formalism reviewed e.g. in Ref. [42].

Let us first recall the definition of the non-Abelian Berry-Wilczek-Zee connection along the Brillouin zone for a set of smooth vectors $\left|u_{n}(\boldsymbol{k})\right\rangle$, $n = 1,...M$:

$$\mathbf{A}_{nm}(\boldsymbol{k}) = \left\langle u_{n}(\boldsymbol{k})\right| \partial_{\boldsymbol{k}} \left|u_{m}(\boldsymbol{k})\right\rangle. \tag{27}$$

The associated Wilson loop operator defined along the path $\mathcal{C}_{j}$ through the Brillouin zone is given by the ordered exponential

$$W_{j} = \overline{\exp}\left(-\int_{\mathcal{C}_{j}} d\boldsymbol{k}\cdot\mathbf{A}(\boldsymbol{k})\right). \tag{28}$$

The topological properties of a generic gapped chiral Hamiltonian are conveniently captured by smooth deformations yielding a flat spectrum $E = \pm 1$. The corresponding Bloch Hamiltonian is then given by

$$H = \begin{pmatrix} 0 & Q(\boldsymbol{k}) \\ Q^{\dagger}(\boldsymbol{k}) & 0 \end{pmatrix}, \tag{29}$$

where $Q(\boldsymbol{k})$ is a nonsingular unitary matrix. Without loss of generality, we write the corresponding eigenstates as

$$\left|u_{\pm n}(\boldsymbol{k})\right\rangle = \frac{1}{\sqrt{2}}\begin{pmatrix} \pm Q(\boldsymbol{k})\left|e_{n}^{B}\right\rangle \\ \left|e_{n}^{B}\right\rangle \end{pmatrix}, \tag{30}$$

where the sign $\pm$ identifies the sign of the eigenvalue $E = \pm 1$ and the normalized vectors $\left|e_{n}^{B}\right\rangle$ form a basis of the Hilbert space of $Q^{\dagger}$. The non-Abelian connection (27) for the negative (resp. positive) energy states then takes the simple form

$$\mathbf{A}_{nm}^{-}(\boldsymbol{k}) = \frac{1}{2}\left\langle e_{n}^{B}\right| Q^{\dagger}(\boldsymbol{k})\partial_{\boldsymbol{k}}Q(\boldsymbol{k})\left|e_{m}^{B}\right\rangle = \mathbf{A}_{nm}^{+}(\boldsymbol{k}). \tag{31}$$

It follows from the definition of the Wilson-loop operator (Eq. (28)) that the intercellular Zak phase for the negative energy bands $\gamma = \gamma^{A} + \gamma^{B}$ is defined in terms of the Wilson loops for the non-Abelian connection $\mathbf{A}^{-}(\boldsymbol{k})$ as

$$\gamma_{j} = -i \ln \det W_{j}^{-}. \tag{32}$$

The quantization of all $d$ intercellular Zak phases then follows from Eqs (28) and (31):

$$\gamma_j = -i \operatorname{tr} \ln \left[ \overline{\exp} \left( -\frac{1}{2} \int_{\mathcal{C}_j} d\boldsymbol{k} \cdot \partial_{\boldsymbol{k}} \ln Q(\boldsymbol{k}) \right) \right] \tag{33}$$

$$= \pi w_j \bmod (2\pi), \tag{34}$$

where the $\bmod (2\pi)$ indetermination stems from the choice of the branch cut of the complex ln function, and where $w_j$ is the standard winding of the chiral Hamiltonian (29):

$$w_j = \frac{i}{4\pi} \int_{\mathcal{C}_j} d\boldsymbol{k} \cdot \operatorname{tr} \left[ \partial_{\boldsymbol{k}} H \mathbb{C} H^{-1} \right] \in \mathbb{Z}, \tag{35}$$

$$= \frac{1}{2\pi i} \int_{\mathcal{C}_j} d\boldsymbol{k} \cdot \operatorname{tr} \left[ Q^{-1} \partial_{\boldsymbol{k}} Q \right]. \tag{36}$$

We therefore conclude that the $d$ Zak phases are topological phases defined modulo $2\pi$.

### B.4 Relating the sublattice Zak phases to the winding of the Bloch Hamiltonian.

We here demonstrate the essential relation given by Eq. (9). To do so, we relate the winding $w_j$ to the sublattice Zak phases by evaluating the trace in Eq. (35) using the eigenstate basis. Noting that $\langle u_n | \partial_{\boldsymbol{k}} H(\boldsymbol{k}) \mathbb{C} H^{-1}(\boldsymbol{k}) | u_n \rangle = -2 \langle u_n | \mathbb{C} \partial_{\boldsymbol{k}} | u_n \rangle$, the winding takes the simple form

$$w_j = -\frac{i}{2\pi} \int_{\mathcal{C}_j} d\boldsymbol{k} \sum_n \langle u_n | \mathbb{C} \partial_{\boldsymbol{k}} | u_n \rangle. \tag{37}$$

Decomposing the chiral operator on the two sublattice projectors $\mathbb{C} = \mathbb{P}^A - \mathbb{P}^B$, yields

$$\pi w_j = \left( \gamma_j^B - \gamma_j^A \right) \in \pi \mathbb{Z}. \tag{38}$$

### B.5 Quantization of the sublattice Zak phases.

Eqs. (34) and (38) shows that both the sum and the difference of the sublattice Zak phases are quantized:

$$\begin{aligned} \gamma_j^A + \gamma_j^B &= \pi w_j + 2\pi m, \quad m \in \mathbb{Z}, \\ \gamma_j^B - \gamma_j^A &= \pi w_j. \end{aligned} \tag{39}$$

It then follows that both sublattice phases $\gamma_j^A$ and $\gamma_j^B$ are integer multiples of $\pi$.

### B.6 How does the winding number of a chiral Bloch Hamiltonian change upon unit cell redefinition?

Starting from a chiral Hamiltonian $\mathcal{H}$, we demonstrate below the relation between the winding numbers associated to the Bloch Hamiltonians constructed from different choices of unit cells, Eq. (11).

The definition of Bloch waves and Bloch Hamiltonians require prescribing a unit cell. Starting with a first choice of a unit cell geometry, say unit cell (1), we can write $H^{(1)}(\boldsymbol{k})$ in the chiral basis as

$$H^{(1)}(\boldsymbol{k}) = \begin{pmatrix} 0 & Q^{(1)} \\ Q^{\dagger(1)} & 0 \end{pmatrix}. \tag{40}$$

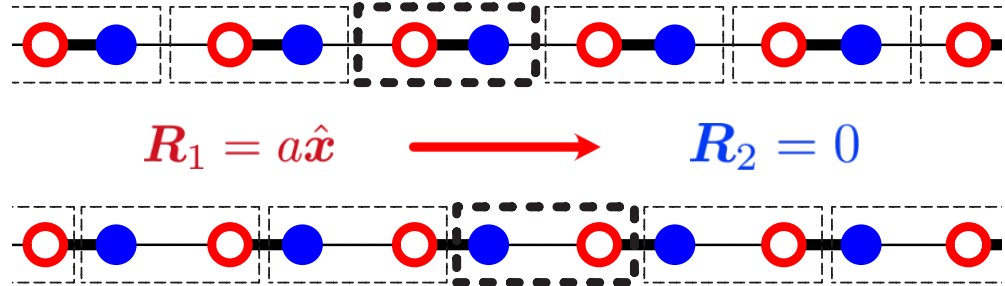

Figure 10: **Unit cell transformation.** We illustrate the definition of the $\mathbf{R}_\alpha$ vectors using the simple example of a SSH chain. For the first atom (empty symbol) $\mathbf{R}_1 = a\hat{\mathbf{x}}$ while $\mathbf{R}_2 = 0$ for the second atom (solid symbol).

Let us now opt for a second choice of unit cell, say choice (2). The Bloch Hamiltonians $H^{(1)}$ and $H^{(2)}$ are then related by a unitary transformation

$$H^{(2)} = U^\dagger H^{(1)} U \,, \tag{41}$$

where the components of the unitary matrix are given by

$$U_{\alpha\beta} = \exp\left(i\mathbf{k}\cdot\mathbf{R}_\alpha^{(12)}\right)\delta_{\alpha\beta}\,, \tag{42}$$

where the $\mathbf{R}_\alpha^{12}$ are the Bravais vectors connecting the position of the atoms in the two unit-cell conventions, see Fig. 10 for a simple illustration. We note that, we have implicitly ignored the trivial redefinitions of the unit cell that reduce to permutations of the site indices. We can then express the winding of $H^{(2)}$ using Eq. (41) in the definition of Eq. (35), which yields

$$w_j^{(2)} = \frac{i}{4\pi}\int_{\mathcal{C}_j} d\mathbf{k}\,\mathrm{tr}\left[\partial_{\mathbf{k}}(UH^{(1)}U^\dagger)\mathbb{C}(UH^{(1)}U^\dagger)^{-1}\right]\,. \tag{43}$$

Expanding the gradient, using the trace cyclic property and noting that $[\mathbb{C}, U] = 0$, we find

$$w_j^{(2)} = w_j^{(1)} - \frac{i}{2\pi}\int_{\mathcal{C}_j} d\mathbf{k}\,\mathrm{tr}\left[\partial_{\mathbf{k}}U\mathbb{C}U^{-1}\right]\,. \tag{44}$$

This equation relates the winding numbers of the two Bloch Hamiltonians to the winding number of the transformation matrix $U$, which is by definition a geometrical quantity independent of $\mathcal{H}$. Using Eq. (42) leads to the remarkable relation which relates the spectral properties of the Hamiltonian to the unit-cell geometry

$$w_j^{(1)} - w_j^{(2)} = \frac{i}{2\pi}\int_{\mathcal{C}_j} d\mathbf{k}\,\mathrm{tr}\left[\partial_{\mathbf{k}}U\mathbb{C}U^{-1}\right] = \frac{1}{a_j}\left(\sum_{\alpha\in A} R_\alpha - \sum_{\alpha\in B} R_\alpha\right)\,. \tag{45}$$

## C  Zero energy flat-band insulators.

We consider a flat-band chiral insulator, defined on a lattice with an non-vanishing chiral charge. In mechanics this situation is readily achieved adding extra bonds to further rigidify an

otherwise isostatic lattice. It is characterized by a finite gap separating positive and negative energy states and by an additional flat band at $E = 0$. In such a phase, there may exist additional zero energy edge states in addition to the bulk zero-energy modes. These edge states are analogous to to the topological edge modes of insulators. Our goal is here to derive a bulk-boundary correspondence for these materials and provide a count of their zero-energy edge states. We will show that this correspondence involves the specific geometry of the eigenstates as opposed to their topology in the case of genuine insulators.

To show this we first derive the expression of the chiral polarization in the presence of a finite bulk chiral charge. Our starting point is Eq. (22), which relates to the chiral polarization of a crystal to the sublattice Zak phases given by Eq. (20):

$$\mathbf{\Pi} \equiv 2\sum_{n<0}\left\langle x^A\right\rangle_{n,R} - \left\langle x^B\right\rangle_{n,R} = \frac{2}{\Omega}\sum_{n<0}\mathbf{\Gamma}^A(n) - \mathbf{\Gamma}^B(n). \tag{46}$$

The sum over all the negative energy bands $n < 0$ is half the sum over the non-zero energy states $n \neq 0$ given by

$$\sum_{n\neq 0}\mathbf{\Gamma}^a(n) = \int_{BZ}\mathrm{d}^d k \sum_{\alpha\in a}\sum_n u^*_{n,\alpha}u_{n,\alpha}r_\alpha + \frac{2}{\Omega^{1/d}}\gamma^a$$

$$= \int_{BZ}\mathrm{d}^d k \sum_{\alpha\in a}\left(1-\sum_{n_0}u^*_{n_0,\alpha}u_{n_0,\alpha}\right)r_\alpha + \frac{2}{\Omega^{1/d}}\gamma^a. \tag{47}$$

In the last line, we single out the role of the bulk zero-energy modes indexed by $n_0$. Using the above expression to evaluate the r.h.s. of Eq. (46), we find an expression similar to Eq. (25) in the main text:

$$\mathbf{\Pi} = (\mathbf{p} - \mathbf{p}_{\mathrm{ZM}}) + \frac{2}{\Omega^{1/d}}\left(\gamma^A - \gamma^B\right). \tag{48}$$

A first noticeable difference with Eq. (25) is a spectral correction to the geometrical polarization stemming from the localized zero-energy bulk modes. This zero-mode polarization is given by

$$\mathbf{p}_{\mathrm{ZM}} = -\int_{BZ}\mathrm{d}^d k \sum_{n_0}\left(\sum_{\alpha\in A} - \sum_{\alpha\in B}\right)u^*_{n_0,\alpha}u_{n_0,\alpha}r_\alpha. \tag{49}$$

Three comments are in order. Firstly, we stress that while the geometrical polarization $\mathbf{p}$ depends on the choice of origin in the presence of an excess of chiral charge, the difference $\mathbf{p} - \mathbf{p}_{\mathrm{ZM}}$, and $\mathbf{\Pi}$, are both independent of the frame's origin. Secondly, unlike in insulators, the difference between the intercellular sublattice Zak phases, $\gamma^A - \gamma^B$ is does not identify with the winding number of the Bloch Hamiltonian. In fact it is not a topological quantity: it continuously depends on the specific couplings of the Hamiltonian. Finally, we point that, by definition, the chiral polarization does not depend on the Bloch convention. A change in the Bloch convention changes the geometrical polarization, the zero-mode polarization, and the intercellular zak phases in such a way that all corrections cancel one another.

Equiped with Eq. (48), we now now turn to the generalization of the bulk boundary correspondence for flat-band insulators. We consider a crystalline material $\mathcal{S}$ terminated by a clean edge $\partial\mathcal{S}$ oriented along the Bravais vector $\mathbf{a}_1$. This edge may host $\mathcal{V}^{\mathrm{NT}}$ non-trivial zero-energy modes, in addition to the (trivial) bulk zero modes associated to the flat band. The edge defines a unit cell that may not be compatible with that of the atomic limit. We can nonetheless extend the edge region such that it matches the unit-cell compatible with the atomic limit (AL). The idea being that $\mathcal{V}^{\mathrm{NT}}$ is fully determined by the additional chiral charge of the edge with respect to that provided by the bulk chiral charge density. Following the same reasoning as

in the main text, this extra chiral charge is given by the difference of geometrical polarization and zero-mode polarization:

$$\mathcal{V}^{\mathrm{NT}} = \mathcal{N}^{\mathcal{B}} \big[ (p_2 - p_{\mathrm{ZM2}})_{\mathrm{AL}} - (p_2 - p_{\mathrm{ZM2}}) \big], \tag{50}$$

where $\mathcal{N}^{\mathcal{B}}$ is the boundary length expressed in units of unit-cell length. The first term is computed in the unit cell compatible with the atomic limit, and the second term is computed in the original unit cell defined by the edge $\partial \mathcal{S}$.

The invariance of the chiral polarization with respect to unit cell transformations allows the connection with the intercellular sublattice Zak phase:

$$\left( p_2 - p_{\mathrm{ZM2}} + \frac{2}{\Omega^{1/d}} \left( \gamma_2^A - \gamma_2^B \right) \right)_{\mathrm{AL}} = p_2 - p_{\mathrm{ZM2}} + \frac{2}{\Omega^{1/d}} \left( \gamma_2^A - \gamma_2^B \right), \tag{51}$$

where AL denotes the terms evaluated in the unit-cell compatible with the atomic limit. All in all, the non-trivial zero-energy content of flat band insulators is given by a formula whhich generalizes Eq. (13):

$$\mathcal{V}^{\mathrm{NT}} = \mathcal{N}^{\mathcal{B}} \frac{2}{\Omega^{1/d}} \big[ (\gamma_2^A - \gamma_2^B) - (\gamma_2^A - \gamma_2^B)_{\mathrm{AL}} \big]. \tag{52}$$

It is worth noting that in the case of genuine insulator, $(\gamma_2^A - \gamma_2^B)_{\mathrm{AL}} = -w_{\mathrm{AL}} = 0$ since it corresponds to the winding number in the unit cell compatible with the AL. Once again the chiral polarization field and its relation with the geometric phases allow us to predict the existence of non-trivial zero-energy modes by observing the local discontinuities of the chiral polarization field at any interface.

## D   Basis of localised states: matrix pencil

Finding a localized basis of the space of negative energy states poses several challenges when working in high-dimensional systems. In one dimension, this is an easy task that can be directly solved by finding the eigenstates of the projected position operator, $P\widehat{X}P$, where

$$P = \sum_{E<0} |\Psi_E(\boldsymbol{r})\rangle \langle \Psi_E(\boldsymbol{r})|, \tag{53}$$

is the projector onto the occupied energy states and $|\Psi_E\rangle$ are the eigenstates of the real space hamiltonian $\mathcal{H}$.

It would be tempting to generalize this approach to two and three dimensional systems to find a common basis for the independent components of the projected position: $P\widehat{R}P \equiv \left( P\widehat{X}P, P\widehat{Y}P, P\widehat{Z}P \right)$. However, in general these components do not commute. As proposed in the seminal work of Marzari and Vanderbilt [66], a workaround consists is computing the set of maximally localized Wannier functions $\{W_n\}$, which minimizes the quadratic spread:

$$\Delta r^2 = \frac{1}{Na^2} \sum_n^N \big[ \langle W_n | (P\widehat{R}P)^2 | W_n \rangle - |\langle W_n | P\widehat{R}P | W_n \rangle|^2 \big], \tag{54}$$

Other minimization functions can be used to define localized states such Foster-Boys or the Edmiston-Ruedenberg criteria [67].

Here, we introduce an alternative method based on the matrix pencil of the projected positions (for a general introduction to matrix pencils see [68] and pages 375 and 461 of [69]. The matrix pencil of two matrices $PXP$ and $PYP$ corresponds to their linear combination $L(\alpha_1, \alpha_2) = \alpha_1 PxP + \alpha_2 PYP$, where $\alpha_i$ are two non-zero real coefficients [69]. When

$[PXP, PYP] \neq 0$, the eigenvectors of the matrix pencil leads to a localised basis, whose spreading is comparable to the standard Maximally Localized Wannier Functions (MLWF) at a much reduced computational cost. To illustrate this result, we study in detail the case of a chiral Halmiltonian on a $10 \times 10$ honeycomb lattice, see Fig. 6. We compare the MLWF and the matrix pencil methods in Fig. 11. We implement MLWF method using a gradient descent protocol [67]. The minimization of $\Omega$ converges slowly as shown in Fig. 11b. We now discuss the case of the matrix pencil method and first note that the diagonalization of $L$ for the two limiting cases $\alpha_1 = 0$ and $\alpha_2 = 0$ corresponds to finding a basis of completely localized states along the $\hat{y}$ and $\hat{x}$ direction, respectively. In the general case where $\alpha_{1,2} \neq 0$, $L$ corresponds to the position operator along the direction $\alpha_1 \hat{x} + \alpha_2 \hat{y}$. It is therefore convenient to parametrize this axis according to its polar angle: $L(\theta) = \cos\theta PXP - \sin\theta PYP$. Fig. 11c shows the spreading functional evaluated at the local basis obtained from $L(\theta)$ for different polar angles. In practice, we achieve a comparable and even better localisation with respect to the MLWF method after 200 iterations. The only values of $\theta$ for which the matrix pencil method is not effective corresponds to the crystallographic directions of the lattice, see Fig. 11d. As these directions are known a priori, we can safely and effectively use the matrix pencil method to compute a set of localized states. The gain in terms of computing time is obvious. Both the diagonalization of $L$ and each minimization step of $\Delta r^2$ have a computational complexity of order $N^2$, where $N$ is the system size. Choosing a value of $\theta$ avoiding the Bravais directions allows us to find a set of localized states in one step using the matrix pencil method. It is also worth noting that this method is unrelated to the chiral symmetry of the Hamiltonians considered in the main text and applies broadly.

We now switch to disordered systems and illustrate the performance of the matrix pencil method in Figs. 11d, e and f. The minimization of $\Delta r^2$ for the MLWF method is more time consuming than in crystals, Fig. 11e. Conversely, the diagonalization time of $L$ remains unchanged. The difference with the ordered case is visible when plotting the spreading function $\Omega$ as a function of $\theta$. The peaks along the crystallographic directions widen, as expected, when disorder increases.

In practice, we compute the $\theta$ average of the chiral polarization associated to a set of localized eigenstates along the $\theta$ directions which yields excellent approximations of the Wannier states, see Fig. 5.

## E    Chiral polarization in amorphous materials.

We have seen that the chiral polarization does not depend on the specifics of the unit cell: it is an intrinsic property of the material. In fact, as we show below, this framework is far more general and we can define the chiral polarization in amorphous solids.

We start by revisiting the definition of the chiral polarization in a crystal given by eq. (22):

$$\Pi \equiv 2 \sum_{n<0} \left\langle x^A \right\rangle_{n,\boldsymbol{R}} - \left\langle x^B \right\rangle_{n,\boldsymbol{R}} . \tag{55}$$

Strictly speaking this polarization is defined at the position $\boldsymbol{R}$. However, the discrete translational invariance of the crystal and by consequence, of the Wannier functions, makes the polarization field homogeneous. We can thus we drop the $\boldsymbol{R}$ indices.

The definition of the Wannier function as the inverse Fourier transform of the Bloch eigenstate cannot be used when dealing with a disordered configuration. Instead, we work with a another set of fully localized functions: the eigenstates of the projected position operator onto

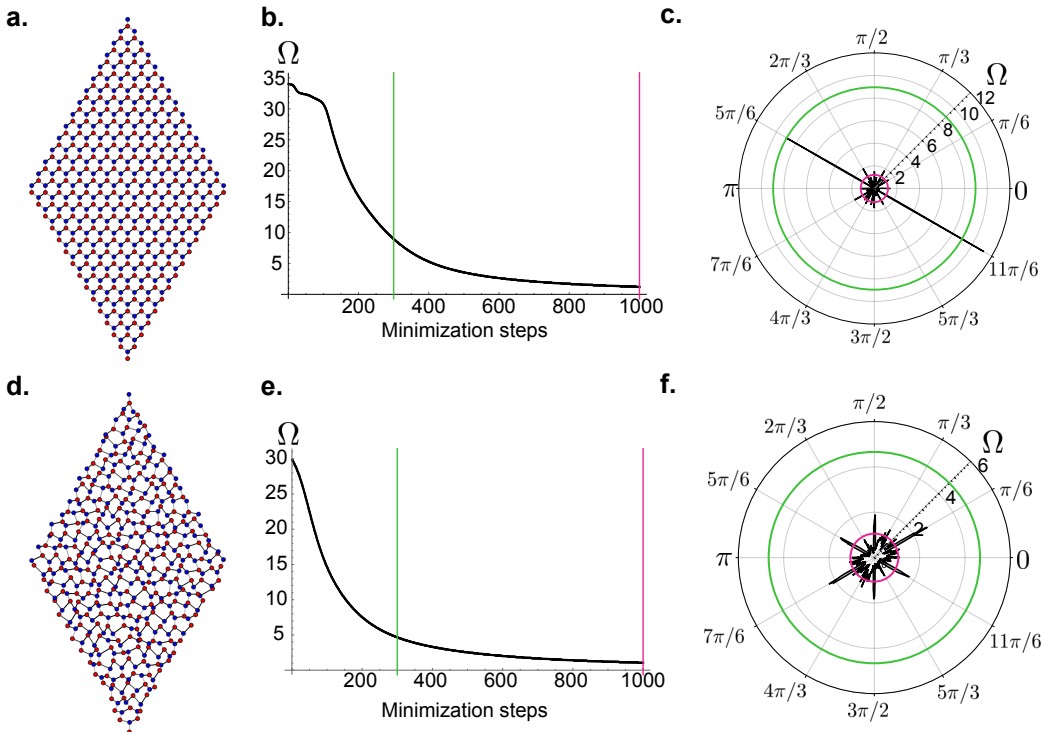

Figure 11: **Comparison between the matrix pencil and the MLWF method a.** Sketch of the frame geometry for a crystalline chiral honeycomb lattice made of $15 \times 15$ unit cells. **b.** Spreading functional as a function of the amount of minimization steps for the determination of maximally localised Wannier functions. After 300 (green) and 1000 (pink) minimization steps, the spreading corresponds to $\Omega_{\text{Wannier}_{300}} \approx 8.9$ and $\Omega_{\text{Wannier}_{1000}} \approx 1.2$, respectively. **c.** Spreading functional as a function of the angle $\theta$ for the localised basis determined from the matrix pencil $L(\theta)$ (black line). In green and pink we show the spreading obtained from the wannier states after 300 and 1000 minimization steps, respectively. Except for a few given directions, notably $\theta = 11\pi/6$ and $\theta = 5\pi/6$, the matrix pencil method gives a more localised basis at a much lower computational cost. **d., e., f.** Same as before applied for a disordered system with $|\delta\boldsymbol{x}|/a = 0.2$, $\epsilon_{\text{D}} = 0.4$, and $\xi = 10a$. (see figs. 6 and 7)

the occupied bands [42]. The projected position operator is given by $P\widehat{X}P$, where

$$P = \sum_{E<0} |\Psi_E(\boldsymbol{r})\rangle \langle\Psi_E(\boldsymbol{r})| , \tag{56}$$

is the projector onto the occupied energy states (not to be confused with the projectors $\mathbb{P}^a$), and the $|\Psi_E\rangle$ are the eigenstates of the real space hamiltonian $\mathcal{H}$. Let us denote the $m^{\text{th}}$ eigenstate of the projected position operator as $\widetilde{W}_m$ (notice that there are as many eigenstates as occupied energy states of the Hamiltonian). This is a localized function around the center given by $\boldsymbol{x}_m = \langle\widetilde{W}_m|\widehat{X}|\widetilde{W}_m\rangle$, similarly to the Wannier centers. Moreover, using each localized function, we can compute the difference of the weighted positions on both sublattices, in other words, the local chiral polarization:

$$\boldsymbol{\Pi}(\boldsymbol{x}_m) = 2\langle\widetilde{W}_m|\mathbb{C}\widehat{X}|\widetilde{W}_m\rangle . \tag{57}$$

In a periodic frame, the eigenstates of the projected position operator reduce to a linear combination of the Wannier functions $W_n$: $\left|\widetilde{W_m}\right\rangle = \sum_n V_{mn} \left|W_n\right\rangle$, with $n < 0$, indicating the occupied energy bands, $V$ a unitary matrix in the energy space, and $V_{mn}$ a diagonal matrix in the position space. We can then rewrite the chiral polarization in eq. (57) as

$$
\begin{aligned}
\Pi(\boldsymbol{x}_m) &= 2\sum_{n,l} \langle W_n| V_{mn}^\dagger \mathbb{C} V_{ml} |W_l\rangle \\
&= 2\sum_{n<0} \langle W_n|\mathbb{C}\widehat{X}|W_n\rangle \,,
\end{aligned}
\tag{58}
$$

where in the last line we used the fact that the $V_{ml}$ commutes with $\mathbb{C}\widehat{X}$ and the unitarity of $V$. As a result, we recover the first expression defined in crystals using the Bloch formalism as given by Eq. (22).

## F  Chiral polarization of time evolved Wannier states.

In Ref. [56], the mean chiral displacement under Hamiltonian dynamics was introduced as a measure of the Zak phase of periodic Hamiltonians in $d = 1$. This quantity characterizes a representation of a Hamiltonian associated to a given unit cell definition, and corresponds to the long-time displacement of an initially fully localized state, measured in units of the unit-cell size. As a consequence, several choices of unit cells were necessary to fully characterize the dynamics of a given (meta)material [55]. The chiral polarization which we extensively use in this article is an intrinsic (meta)material property, unlike the mean chiral displacement and the skew polarization. It is defined in real space, and does not rely on any underlying frame periodicity, Eq. (5) In the specific case of periodic frames $\Pi$ crucially resolves the chiral imbalance of wave packets with a sub-unit-cell resolution.

In this method section, we show how $\Pi$ relates to the dynamics of a maximaly localized Wannier state spreading in the bulk of a chiral crystal. To do so we consider the time evolution of a wave function $|\psi_n(t)\rangle = U(t)\left|W_{n,\boldsymbol{R}}\right\rangle$ starting from a of a Wannier state in band $n$, initially localized at $\boldsymbol{R}$, with an evolution operator $U(t) = \exp(-iHt)$. Using the notations introduced in Eq. (16), the position at time $t$ is given by

$$
\langle \boldsymbol{x}|\widehat{X}|\psi_n(t)\rangle = \Omega^{-1} \int_{\mathrm{BZ}} \mathrm{d}^d \boldsymbol{k}\, \boldsymbol{x}\, e^{i\boldsymbol{k}\cdot(\boldsymbol{x}-\boldsymbol{R})} e^{-iE_n(\boldsymbol{k})t} \varphi_{n,\boldsymbol{k}}(\boldsymbol{x})
\tag{59}
$$

$$
= \Omega^{-1} \int_{\mathrm{BZ}} \mathrm{d}^d \boldsymbol{k}\, e^{-i\boldsymbol{k}\cdot\boldsymbol{R}} \left[ e^{i\boldsymbol{k}\cdot\boldsymbol{x}} \left(\boldsymbol{R} + \boldsymbol{v}_n(\boldsymbol{k})t + i\partial_{\boldsymbol{k}}\right) \right] \varphi_{n,\boldsymbol{k}}(\boldsymbol{x}),
\tag{60}
$$

where $\boldsymbol{v}_n(\boldsymbol{k}) = \partial_{\boldsymbol{k}} E_n(\boldsymbol{k})$ is the group velocity in the energy band $n$. We can also generalize Eq. (21) to define the instantaneous average positions restricted to the $a = A, B$ sublattices which read

$$
\langle \boldsymbol{x}^a(t)\rangle_{n,\boldsymbol{R}} \equiv \langle \psi_n(t)|\mathbb{P}^a \hat{X} \mathbb{P}^a|\psi_n(t)\rangle
\tag{61}
$$

$$
= \oint_{\boldsymbol{k}} \frac{1}{2}(\boldsymbol{R} + \boldsymbol{v}_n(\boldsymbol{k})t) + \frac{1}{\Omega}\boldsymbol{\Gamma}_{\mathrm{Zak}}^a(n)
\tag{62}
$$

$$
= \frac{1}{2}\boldsymbol{R} + \frac{1}{\Omega}\boldsymbol{\Gamma}_{\mathrm{Zak}}^a(n)
\tag{63}
$$

$$
= \langle \boldsymbol{x}^a(t=0)\rangle_{n,\boldsymbol{R}} \,.
\tag{64}
$$

This result indicates that the chiral polarization of each Wannier state is a stationary quantity although they all evolve in time . When summed over (half of) the spectrum, we recover the static definition of the chiral polarization

$$\mathbf{\Pi}(t) = \mathrm{Tr}(U^{-1}(t)\mathbb{C}\hat{X}U(t)) \tag{65}$$

$$= \sum_n \langle \psi_n(t)|\mathbb{C}\hat{X}|\psi_n(t)\rangle \tag{66}$$

$$= 2\sum_{n<0} \left\langle \boldsymbol{x}^A(t)\right\rangle_{n,\boldsymbol{R}} - \left\langle \boldsymbol{x}^B(t)\right\rangle_{n,\boldsymbol{R}}$$

$$= \frac{2}{\Omega}\sum_{n<0} \mathbf{\Gamma}^A_{\mathrm{Zak}}(n) - \mathbf{\Gamma}^B_{\mathrm{Zak}}(n). \tag{67}$$

We note that the trace operation in Eq. (65) can be evaluated using any basis of the Hilbert space, such as the ensemble of states fully localized on the $A$ and $B$ sites.

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
