# Peer review of "Geometry and Topology Tango in Ordered and Amorphous Chiral Matter"

_SciPost Physics, doi:SciPost Phys. 12, 038 (2022)_

## Round 1 · Referee Report · Daniel Varjas (Referee 1) · 2021-8-3

Strengths

1) The manuscript introduces the concept of (local) chiral polarization, and relates domain walls in the chiral polarization to the presence of zero modes. This novel approach is of great interest to the topological materials and topological mechanics community.

2) The manuscript demonstrates the power of these methods in systems with strong disorder, including amorphous materials.

Weaknesses

1) The pencil matrix method to estimate the Wannier centers without performing the costly optimization of the spread functional is of great interest to the community. However, whether this method is generally applicable is a highly nontrivial question, that the manuscript does not address, beyond comparing the results in a few specific cases and finding good agreement. Besides a strong argument for the method's applicability missing (with reference 51 to an entire book being too vague), it is also unclear what exactly the method is. I thank the authors for making their Mathematica code available, however, this code only calculates results for a single choice of α's, while the results in the manuscript are obtained by averaging over an ensemble. What is the distribution the α's are drawn from? How are the position eigenstates for various α values matched while averaging? Considering that one choice of α's corresponds to calculating the projected position eigenvalues along one randomly chosen axis, it is not obvious which eigenstate corresponds to the same Wannier center in a large sample for different choices of α's.

2) The manuscript considers systems with a chiral symmetry of the sublattice type, i.e. not every site contains an equal number of even and odd orbitals under chiral symmetry, but the chiral symmetry arises from the bipartite nature of the lattice. I am not convinced of the physical relevance of this case for amorphous matter as studied in sec 5.3, as naturally occurring amorphous lattices are rarely strictly bipartite.

Report

The manuscript meets the standards of the journal by opening a new pathway in an existing research direction, with clear potential for multipronged follow-up work. The main results in the manuscript are correct, and of significant interest to the topological condensed matter community. However, some of the results are not presented in a clear, or detailed enough manner, and I request changes before recommending publication, see the Weaknesses and Requested changes sections.

Requested changes

1) I ask the authors to address the criticism in the "Weaknesses" section of the report.

2) There seem to be several typos related to prefactors included in the definition of various quantities, and the gauge ambiguities of them. For example in (39) m∈πZ is clearly a typo and should be m∈Z. Moreover, some definitions seem to be inconsistent, for example using the definition (4) of γ^A/B, going from (37) to (38) a factor of 2 seems missing. Overall it is quite confusing that the definitions in Appendix A.3, B.1 and B.2 contain extra factors of BZ volume compared to those used in the main text. It is often hard to decide whether the k integrals are over the volume of the BZ, or along 1D lines. I ask the authors to unify the notation and carefully check for typos.

3) There are several references to "Methods", these should be replaced with references to specific Appendices.

4) The term "atomic limit" is used throughout the manuscript without defining it. While it is clear from the context for experts on the topic, this terminology may be confusing, as the atomic limits in question are in fact perfectly dimerized limits. I ask the authors to clarify the terminology in the manuscript. It is also worth noting, that in more complicated systems, with more types of (further neighbour) hoppings, it may be less obvious what the atomic limit of a given system is, it would be useful to give a more general definition.

  • validity: high
  • significance: high
  • originality: high
  • clarity: good
  • formatting: perfect
  • grammar: perfect

Author:  Marcelo Guzmán  on 2021-09-23  [id 1774]

(in reply to Report 1 by Daniel Varjas on 2021-08-03)
Category:
answer to question

Detailed answers to your questions and comments:

1) The pencil matrix method to estimate the Wannier centers without performing the costly optimization of the spread functional is of great interest to the community. However, whether this method is generally applicable is a highly nontrivial question, that the manuscript does not address, beyond comparing the results in a few specific cases and finding good agreement. Besides a strong argument for the method's applicability missing (with reference 51 to an entire book being too vague), it is also unclear what exactly the method is. I thank the authors for making their Mathematica code available, however, this code only calculates results for a single choice of α's, while the results in the manuscript are obtained by averaging over an ensemble. What is the distribution the α's are drawn from? How are the position eigenstates for various α values matched while averaging? Considering that one choice of α's corresponds to calculating the projected position eigenvalues along one randomly chosen axis, it is not obvious which eigenstate corresponds to the same Wannier center in a large sample for different choices of α's.

We do agree with you. We did not provide enough details about this original method in our first version of the main text. We thank you for suggesting a more detailed presentation of this new method. Following your advice, we have included a detailed discussion of the matrix pencil method in Appendix D and added a new Figure (Fig. 11). This discussion shows how to suitably choose the parameter $\alpha$ in periodic lattices and how the computational cost of this method compares with the conventional computation of Maximally Localized Wannier States.

The Mathematica file we upload was indeed incomplete. We have corrected this unfortunate mistake. Thank you for having pointed it out.

2) The manuscript considers systems with a chiral symmetry of the sublattice type, i.e. not every site contains an equal number of even and odd orbitals under chiral symmetry, but the chiral symmetry arises from the bipartite nature of the lattice. I am not convinced of the physical relevance of this case for amorphous matter as studied in sec 5.3, as naturally occurring amorphous lattices are rarely strictly bipartite.

We agree with you in the context of solid state physics, where chiral symmetry typically emerges at low energy. However, we stress that photonic metamaterial can be chiral by design and that all mechanical structures enjoy an intrinsic chiral symmetry as first revealed by Kane and Lubensky (Nature Physics 2014). Remarkably this features holds both in homogeneous and heterogeneous mechanical metamaterials. Each positional degree of freedom (A site) is only connected to a stress degrees of freedom (B site). We stress on this crucial point in the main text (Introduction of Section 5 line 285-291)

  1. There seem to be several typos related to prefactors included in the definition of various quantities, and the gauge ambiguities of them. For example in (39) m∈πZ is clearly a typo and should be m∈Z. Moreover, some definitions seem to be inconsistent, for example using the definition (4) of γ^A/B, going from (37) to (38) a factor of 2 seems missing. Overall it is quite confusing that the definitions in Appendix A.3, B.1 and B.2 contain extra factors of BZ volume compared to those used in the main text. It is often hard to decide whether the k integrals are over the volume of the BZ, or along 1D lines. I ask the authors to unify the notation and carefully check for typos.

Thank you for you thorough reading of our manuscript. We have corrected all the typos you found and have carefully double checked all of our notations:

“For example in (39) m∈πZ is clearly a typo and should be m∈Z.”: 
We corrected it.

“Moreover, some definitions seem to be inconsistent, for example using the definition (4) of γ^A/B, going from (37) to (38) a factor of 2 seems missing.”:
There is no inconsistency in this definition. (4) corresponds to the sublattice Zak phase of one energy band (indexed by n). However, from (37) to (38) we use the sum over all the bands, giving twice the total sub lattice Zak phase.
 “Overall it is quite confusing that the definitions in Appendix A.3, B.1 and B.2 contain extra factors of BZ volume compared to those used in the main text. It is often hard to decide whether the k integrals are over the volume of the BZ, or along 1D lines. I ask the authors to unify the notation and carefully check for typos.”: 
We did check for typos and misprints to make our manuscript and Appendix as clear as possible. To avoid any possible confusion we now use a consistent notation for the integrals over the BZ and we chose this notation as distinct as possible from the contour integrals defining the Zak phases.

  1. There are several references to "Methods", these should be replaced with references to specific Appendices.

We have done so.

  1. The term "atomic limit" is used throughout the manuscript without defining it. While it is clear from the context for experts on the topic, this terminology may be confusing, as the atomic limits in question are in fact perfectly dimerized limits. I ask the authors to clarify the terminology in the manuscript. It is also worth noting, that in more complicated systems, with more types of (further neighbour) hoppings, it may be less obvious what the atomic limit of a given system is, it would be useful to give a more general definition.

Following your suggestion, we have added a formal and a practical definition of the atomic limit in Section 3:

« We recall that the atomic limit of a material corresponds to a smooth deformation of the couplings to separate the energy scales so that so that the Wannier functions are exponentially localized, and respect the symmetries of the crystal~\cite{Bradlyn2017}. in practice, it consists in choosing a unit cell including the strongest couplings. » (We also refer to the work of Bradley et al.)

Attachment:

---

## Round 2 · Referee Report · Anonymous · 2021-11-18

Strengths
1 - The chiral polarization has the potential of being a useful quantity in determining the topological properties of disordered and/or amorphous systems.
2 - The presentation is well-structured, with simple examples helping the reader gain an intuitive understanding of the authors' results.
Weaknesses
1 - I believe that figurative/metaphorical language should be avoided in scientific publications.
2 - The relation between this work and previous research could be made more explicit.
Report
The authors introduce a quantity called the "chiral polarization." It is a real space quantity which helps to clarify the interplay between topology and lattice geometry in chiral-symmetric systems, and seems useful as a real-space marker that characterizes different domains in uniform, disordered, and/or amorphous systems.
The presentation is well structured. The quantities used by the authors as well as their results are introduced in a step-by-step fashion, with examples based on simple systems helping the reader to gain an intuitive understanding of the authors' work. In my opinion, this work does meet the Scipost acceptance criteria. Specifically, it opens a new pathway in an existing research direction, with clear potential for multipronged follow-up work (https://scipost.org/SciPostPhys/about).
Before publishing this work, however, I would urge the authors to address the following points.
Requested changes
Main points:
1 - My main criticism of this work concerns the way in which the authors relate their findings to the existing body of research. I believe that this relation should be made more accurate and explicit, so as to avoid potentially misleading the readers. For example, on line 123 the authors write that the chiral polarization is "seemingly identical" to the skew polarization and the mean chiral displacement. Either they are identical, or they are not.
My current understanding (which may be wrong, please correct me if that is the case) is that the chiral polarization is in fact mathematically identical to the mean chiral displacement (MCD). They are the same quantity. One of the main novelties of this work, as far as I can understand, is that the authors compute the MCD using a specific set of states, the maximally localized states. This means they can use the MCD as a real-space indicator, which is useful in describing the topological domains of large disordered and/or amorphous systems. This is indeed an important result, but I wish it would be related to previous research in as accurate and explicit a way as possible.
Beyond line 123, I have found similar expressions in other parts of the paper. On line 328 the authors compare their results to the mean chiral displacement by using the words: "out of reach of conventional chiral displacement characterizations" and cite Ref. 52. This reference is a combined experimental/numerics paper measuring the MCD. Do the authors mean that the MCD is fundamentally unable to produce the same characterizations? This would be confusing to me, since their chiral polarization is mathematically identical to the MCD, as far as I can understand. Do they instead mean that the results of Ref. 52 suffer from finite-size and finite-time effects? If so, then please state this explicitly.
On line 445, they state "our protocol is close to the chiral displacement method." On line 747, they compare the chiral polarization with the MCD protocols by saying that they are "seemingly similar." To me it seems the authors' time-evolution protocol is literally identical to the MCD method. If so, please state this explcitly.
2 - In Fig. 6, the authors show that the chiral polarization remains a good indicator even when the sites of the regular lattice are shifted from their original positions. To me, it does make sense that the zero modes are robust to such a change, since the Hamiltonian matrix istelf remains constant. However, it is not obvious to me whether the robustness of the chiral polarization will persist for larger disorder strength. For example, if setting $|\delta x|/a=10$, the zero modes will clearly still be there, but will the chiral polarization be able to tell? I realize that such large shifts would probably require a large tight-binding model, but this seems achievable given Fig. 8.
3 - On line 369, the authors state that "The phase boundaries are then readily detected by jumps of the chiral-polarization vector". Can this statement be made quantitative? How high and how sharp should these jumps be before one can conclude that a phase boundary exists?
Minor points:
1 - I strongly believe that metaphorical/figurative language has no place in scientific publications. I urge the authors to remove and/or rephrase the following:
"tango" in the title, "spread frantically" on lines 53-54, "intimate interplay" on line 87, "intimate relation" on line 166, "illuminate the very definition" on line 205, "illuminates the geometrical implication" on line 278, "the frame topology and the frame geometry conspire" on lines 450-451, "The subtle tango" on line 458
2 - There are typos in the reference list. Some of the reference titles should contain capitalized words, such as Berry and Wannier. Further, journal abbreviations and formatting is inconsistent across the reference list. Some references are missing links.
3 - I found several minor typos throughout the paper text.
4 - There are two references to movies/videos, but I could not find them in the supplemental material or in the reference list.
Author: Marcelo Guzmán on 2021-11-23 [id 1968]
(in reply to Report 1 on 2021-11-18)
Dear reviewer, We thank you for your encouraging report. We have taken your very useful comments to further clarify the difference between the Mean Chiral Display event and the chiral polarisation. The two quantities are mathematically different and convey two different informations on chiral materials. This was indeed an important point to address. We have also taken care of polishing the main text and correcting some unfortunate typos.
We hope you will find this revised version suitable for publication in SciPost. We provide detailed answers to your questions and comments below Kind regards.
Main points:
I. We thank you for prompting us to clarify the essential difference between our chiral polarisation and the Mean Chiral Displacement (MCD). This point is indeed important and we set out to clarify it. We have made our statements clearer and added a paragraph to discuss quantitively the relation between the Chiral Polarisation and the MCD. The short answer to your question is that the chiral polarisation and the MCD are mathematically different. Let us detail below why this is so.
Using the notations of our manuscript, and focusing on the one dimensional case for clarity, the MCD is defined as $2\left<\mathbb C x_{\text{UC}}\right>$, with $x_{\text{UC}}$ the position of the unit cell. In contrast, the chiral polarisation $\Pi$ is defined by the exact position operator, $2\left<\mathbb C x\right>$. The MCD is defined only in terms of the position on a Bravais lattice, while the chiral polarisation involves the actual position in continuous space. Indeed considering only $x_{\text{UC}}$ amounts to resolve the positions of the atoms only with an accuracy given by the size of a unit cell. The difference is not merely technical, the two quantities are genuinely different. As explained below this difference has fundamental and practical consequences. The MCD explicitly depends on $x_{\text{UC}}$, as a consequence the MCD depends on the choice of unit cell: different choices of unit cells give different MCD values. This is also patent in the dynamical measurements: the MCD converges to the winding number, an integer which is known to be unit-cell dependent, as illustrated in Figure 2. By contrast, the chiral polarisation is free from this ambiguity since it does not require any choice of unit cell. $\Pi$ is a material property. $\Pi$ does not depend on the lattice representation of the material.
To stress this difference, let us decompose explicitly the position operator into: $x=x_{\text{UC}}+\delta x$, the first one being the position operator at the scale of the unit cell (which depends on the chosen convention of unit cell). The chiral polarisation is then a sum of two contributions (spectral and geometrical) as shown in Equations (5) and (10) in the manuscript. The spectral part is the MCD, yet it must be complemented by the geometrical part, the geometrical polarisation, to define $\Pi$, the material property, measurable and thus independent on the lattice conventions.
Regarding the dynamical protocol. The method we introduce also differs from the one defined in ref. (38). The difference between the two protocols echoes the intrinsic difference between the two quantities. The detection of the chiral polarisation relies on a single measurement exciting two sites in a unit cell. In contrast, to detect an invariant with the MCD method, two independent measurements must be performed, one on each unit cell convention, or in the jargon of Floquet systems, two time-frames.
We thank you again for your comment which helped us to better distinguish our work from earlier literature.
Specific changes in the main text:
Line 123: We changed “Although seemingly identical to…” to “This definition differs from…”
Line 128: We added “This difference is simply explained by considering the mean chiral displacement (MCD) defined on a 1D lattice given a definition of a unit cell. It is defined per wavepacket $\psi$ as ${\rm MCD}=\expval{\mathbb C x_{\text{UC}}}{\psi}$, with $x_{\text{UC}}$ being the position operator at the scale of the unit cell. By definition, $x_{\text{UC}}$ is defined as an integer multiple of the length $a$ of the unit cell. In contrast, equation~\eqref{eq:defPolarization} depends on the actual position of the sites: $x=x_{\text{UC}}+\delta x$, where $\delta x$ is a sublattice correction to the unit-cell position .”
Line 444: we changed the last paragraph: “As a last comment we stress that although our protocol is close to the chiral displacement method….” to “As a last comment we stress that our protocol differs from the chiral displacement method introduced and used in~\cite{cardano2017detection,maffei2018topological,st2020measuring,d2020bulk}. The mean chiral displacement depends on the unit-cell convention. As a consequence, to probe the topology of 1D systems, conventional MCD protocols require two independent measurement protocols. They effectively correspond to measuring the mean chiral displacement given two possible unit cell choices. A topological invariant is then defined by the difference between the two measurements. The Chiral polarization method, which we introduce provides a one-step characterization of the topology of a chiral phase. »
line 747: We changed “Although seemingly similar in its formal definition…” to “The chiral polarization which we extensively use in this article is an intrinsic (meta)material property , unlike the mean chiral displacement and the skew polarization. It is defined in real space,…”
II. We would like to stress that the chiral polarization is not only a property of the tight-binding Hamiltonian. Indeed the geometrical part of $\Pi$ depends on the actual positions of the interacting sites. By studying the behavior of the chiral polarization when the sites are displaced but the couplings are kept constant, we question the possibility to extract the topological component of the chiral polarization, embedded in the Hamiltonian, from the full chiral polarization. We discuss the robustness and the limitation of our method in Figure 6 where we show that the method remains valid up to $\delta x/a=1$. When $\delta x/a=10$ the chiral polarisation field would become highly heterogenous with no global domain reflecting the fact that we could not define a line of zero mode any more. The sites hosting the zero modes would form a scattered set of points in this hypothetical case.
III. The jumps associated to a phase boundary in the crystalline case corresponds to a Bravais lattice vector. Hence the minimal jump is of the order of the lattice spacing a. In the disordered case, provided the relative mean displacement <(u_i - u_i+1)^2> is small compared with the original lattice spacing a, we expect the jump to remain of the order of this lattice spacing a. The above condition remains valid in a Bragg glass phase. Deep in an amorphous phase where this criterion becomes invalid, we expect the jump of chiral polarization to become smaller than a and harder to detect without any further knowledge of the spatial structure.
Minor points:
-
We are sorry that you did not appreciate the wording of our manuscript. However, we respectfully disagree with you on this specific point. We do not resort to metaphorical language to refer to physical quantities and concepts. We have asked a native English speaker to check whether the wording was grammatically correct and not misleading and made the appropriate corrections when needed.
-
Thank you for pointing this out. We have double checked the references.
-
We set out to correct typos and misprints and grammar mistakes.
-
The Scipost server did not allow to upload videos. We have made them available on a new version of the Zenodo repository: https://doi.org/10.5281/zenodo.5721386.
Author: Marcelo Guzmán on 2021-12-02 [id 2002]
(in reply to Marcelo Guzmán on 2021-11-23 [id 1968])
Dear referee,
As suggested by the editor we have followed your recommendation and have changed some of the wording in the main text. We hope you and the editor will find the style of the manuscript suitable for publication in SciPost.
Specific changes:
Line 53-54:
“… spread frantically across…”
to
“spanned ”
Line 87:
“… intimate interplay…”
to
“…interplay…”
Line 166
“… intimate relation…”
to
“…relation…”
Line 205:
“… to illuminate the very definition…”
to
”… to clarify the definition…”
Line 278:
“… illuminates the geometrical implication…”
to
“…reveals the geometrical implication…”
Line 450:
“… the frame topology and the frame geometry conspire…”
to
“… the frame topology and the frame geometry act together …”
Line 458:
“This subtle tango…”
to
“This interplay…”
Kind regards
Marcelo, Denis and David
Author: Marcelo Guzmán on 2021-11-29 [id 1989]
(in reply to Report 2 by Daniel Varjas on 2021-11-26)Dear reviewer,
We thank you for your encouraging report. We have taken care of the requested changes and we hope you will find this revised version suitable for publication in SciPost.
A detailed answer can be found below
Kind regards.
Requested changes:
We thank you for this remark. We have now properly referenced the Appendix D in which we detail and quantitatively validate the method. We have added a new reference focused solely on matrix pencils and indicated the relevant pages of the general book on matrix computations (previous reference [58]). In addition, we have moved reference [58] to the Appendix D to avoid any confusion.
Specific changes:
Line 319:
We have changed “…taking advantge of the so-called pencil-matrix method [51].”
To
“…taking advantage of the matrix pencil method detailed in Appendix D.”
Line 678:
We have changed “Here, we introduce an alternative method based on the matrix pencil of the projected positions.”
To
“Here, we introduce an alternative method based on the matrix pencil of the projected positions (for a general introduction to matrix pencils see [68] and pages 375 and 461 of [69] ).”

---

## Round 2 · Referee Report · Daniel Varjas · 2021-11-26

Strengths
The authors have addressed most of my previous comments to my satisfaction.
1) The clarity of the presentation of the pencil-matrix method, as an approximation method to find maximally localized Wannier sates, was greatly improved by the addition of Appendix D and Fig. 11. The limits of applicability of this method are still unclear, and poses an interesting research question.
2) The definition of "atomic limit" was also clarified.
Weaknesses
1) Presentation of the pencil matrix method is still somewhat confusing, see requested changes.
Report
First of all, let me apologise to the Authors and the Editor for the slow report.
I find that the authors sufficiently addressed my criticism in the first report. Besides, I generally agree with the criticism of Anonymous Reviewer, and find that the changes promised by the Authors should be sufficient to address them.
I recommend publication after the changes requested by Anonymous Reviewer and myself are implemented.
Requested changes
1) The citation in "...taking advantge (sic!) of the so-called pencil-matrix method [51]" still seems misleading. A cursory reading of the book didn't illuminate anything about this method beyond the definition of the term "pencil matrix". If the book contains anything relevant beyond the definition, please cite by section or page number. If not, please rephrase this sentence in a way that avoids giving the impression that this method was already introduced in the cited book. A reference to Appendix D should also be included in the main text.

---

## Round 2 · Author Response

List of changes
We have included a detailed discussion of the matrix pencil method in Appendix D and added a new Figure (Fig. 11).
We have added a formal and a practical definition of the atomic limit in Section 3.
We have corrected some typos.

---

## Round 2 · List of Changes

We have included a detailed discussion of the matrix pencil method in Appendix D and added a new Figure (Fig. 11).
We have added a formal and a practical definition of the atomic limit in Section 3.
We have corrected some typos.

---

## Round 3 · Referee Report · Anonymous (Referee 2) · 2021-12-3

Report

First of all, I would like to thank the authors for correcting me and clarifying my confusion concerning the difference between the chiral polarization and the mean chiral displacement.

The authors have addressed all of the points I had raised during the previous round. I have no further comments and I believe the work should be published as is.

---

## Round 3 · Author Response

Resubmission following the suggestions of the referees and the editor.

---

## Round 3 · List of Changes

Line 123:
We changed “Although seemingly identical to…” to “This definition differs from…”

Line 128:
We added “This difference is simply explained by considering the mean chiral displacement (MCD) defined on a 1D lattice given a definition of a unit cell. It is defined per wavepacket $\psi$ as ${\rm MCD}=\expval{\mathbb C x_{\text{UC}}}{\psi}$, with $x_{\text{UC}}$ being the position operator at the scale of the unit cell. By definition, $x_{\text{UC}}$ is defined as an integer multiple of the length $a$ of the unit cell. In contrast, equation~\eqref{eq:defPolarization} depends on the actual position of the sites: $x=x_{\text{UC}}+\delta x$, where $\delta x$ is a sublattice correction to the unit-cell position .”

Line 444:
we changed the last paragraph: “As a last comment we stress that although our protocol is close to the chiral displacement method….” to
“As a last comment we stress that our protocol differs from the chiral displacement method introduced and used in~\cite{cardano2017detection,maffei2018topological,st2020measuring,d2020bulk}. The mean chiral displacement depends on the unit-cell convention. As a consequence, to probe the topology of 1D systems, conventional MCD protocols require two independent measurement protocols. They effectively correspond to measuring the mean chiral displacement given two possible unit cell choices. A topological invariant is then defined by the difference between the two measurements. The Chiral polarization method, which we introduce provides a one-step characterization of the topology of a chiral phase. »

line 747:
We changed “Although seemingly similar in its formal definition…” to “The chiral polarization which we extensively use in this article is an intrinsic (meta)material property , unlike the mean chiral displacement and the skew polarization. It is defined in real space,…”

Line 319:
We have changed “…taking advantge of the so-called pencil-matrix method [51].”
To
“…taking advantage of the matrix pencil method detailed in Appendix D.”

Line 678:
We have changed “Here, we introduce an alternative method based on the matrix pencil of the projected positions.”
To
“Here, we introduce an alternative method based on the matrix pencil of the projected positions (for a general introduction to matrix pencils see [68] and pages 375 and 461 of [69] ).”

Requested language modifications of Anonymous referee:

Line 53-54:
“… spread frantically across…”
to
“spanned ”

Line 87:
“… intimate interplay…”
to
“…interplay…”

Line 166
“… intimate relation…”
to
“…relation…”

Line 205:
“… to illuminate the very definition…”
to
”… to clarify the definition…”

Line 278:
“… illuminates the geometrical implication…”
to
“…reveals the geometrical implication…”

Line 450:
“… the frame topology and the frame geometry conspire…”
to
“… the frame topology and the frame geometry act together …”

Line 458:
“This subtle tango…”
to
“This interplay…”

---

## Editorial Decision

published